# Unbiased Stochastic Optimization for Gaussian Processes on Finite Dimensional RKHS

**Neta Shoham**                                            *shohamne@mail.tau.ac.il*
*School of Mathematical Sciences*
*Tel Aviv University*
*Tel Aviv, Israel*

**Haim Avron**                                             *haimav@tauex.tau.ac.il*
*School of Mathematical Sciences*
*Tel Aviv University*
*Tel Aviv, Israel*

**Reviewed on OpenReview:** *https://openreview.net/forum?id=wDCulUZla4*

## Abstract

Current methods for stochastic hyperparameter learning in *Gaussian Processes* (GPs) rely on approximations, such as computing biased stochastic gradients or using inducing points in stochastic variational inference. However, when using such methods, we are not guaranteed to converge to a stationary point of the true marginal likelihood. In this work, we propose algorithms for exact stochastic inference of GPs with kernels that induce a Reproducing Kernel Hilbert Space (RKHS) of moderate finite dimension. Our approach can also be extended to infinite dimensional RKHSs at the cost of forgoing exactness. Both for finite and infinite dimensional RKHSs, our method achieves better experimental results than existing methods when memory resources limit the feasible batch size and the possible number of inducing points.

## 1  Introduction

Gaussian Processes (GPs) provide a powerful probabilistic framework that has been applied to a wide range of learning applications, such as multi-task learning (Alvarez et al., 2012; Liu et al., 2018a), active learning (Liu et al., 2018b), semi-supervised learning (Jean et al., 2018), and reinforcement learning (Srinivas et al., 2010; Shahriari et al., 2015). These successes can be attributed to the natural way in which uncertainty is incorporated into the predictions via a Bayesian interpretation. However, hyperparameter learning scales poorly. For a general covariance function (kernel function in the parlance of kernel methods), the computational cost grows as $O(n^3)$, where $n$ denotes the number of samples, and the storage resources grow as $O(n^2)$. As a consequence, approximations are required for any modern application that involves GPs and big data.

It is therefore unsurprising that much research effort has been devoted to approximate methods for learning GPs. For a comprehensive overview of this research, we refer the reader to the work of Liu et al. (2020). The most expensive operation in training GPs is the inversion of the $n \times n$ kernel matrix, which is required during the maximization of the marginal likelihood and for the computation of the posterior of the responses. As a result, much research has been devoted to providing a cheap approximation of the kernel matrix.

Broadly speaking, approximations of GPs can be split into *data dependent* and *data independent* methods. Data dependent approximations usually come with valuable probabilistic interpretations. An important work in this line of research is the seminal work of Quinonero-Candela & Rasmussen (2005), which provides a unified view on previous work by using the concept of inducing points. Since then quite a few follow-up

works have used inducing points, while exploiting the tool of variational inference as a theoretical platform (Titsias, 2009; Nguyen et al., 2014; Wilson & Nickisch, 2015; Zhao & Sun, 2016). This approach is closely related to the Nyström approximation of kernel matrices (Zhao & Sun, 2016).

In contrast, data independent methods rely on approximating the kernel function itself. Typically, it is approximated by an inner product between low dimensional vectors (Rahimi & Recht, 2007; Yang et al., 2015; Shustin & Avron, 2021).

Approximating the inverse of the kernel matrix is enough for frequentist kernel methods such as Kernel Ridge Regression. However, in order to harness the full power of Bayesian kernel methods such as *Gaussian Process Regression* (GPR), we also need to be able to maximize the marginal likelihood; for that, an approximation of the inverse of the kernel matrix is not enough due to an additional log-determinant term. This is especially true in cases where the covariance function depends on a large number of parameters, e.g., evaluating the covariance function involves a forward pass of a deep neural network (Wilson et al., 2016b; Calandra et al., 2016; Wilson et al., 2016a). In such cases, it is important to be able to use stochastic optimization based on mini-batches; otherwise, the cost of making a pass over the entire dataset typically proves too expensive. Moreover, it is well appreciated in the literature that when the model involves deep neural networks, i.e., when the covariance function is defined by a neural network, it is crucial to use stochastic gradients, as they enable more efficient optimization and better generalization (Goodfellow et al., 2016).

A key difficulty is that the log-determinant term is global. In the finite-feature setting considered below, the feature covariance can be written as a sum of per-sample contributions, but the log determinant of this sum does not decompose into a sum of per-sample losses. Therefore, the naive mini-batch objective obtained by replacing the full dataset with a mini-batch generally yields biased gradients of the full marginal likelihood.

Currently, two main approaches are used for stochastic hyperparameter learning in GPs. The first relies on the inducing-points framework and employs *Stochastic Variational Inference* (SVI) (Hoffman et al., 2013; Hensman et al., 2013; Hoang et al., 2015; Wilson et al., 2016a). The second approach is more direct: it is based on computing stochastic batches while ignoring the fact that they provide only biased estimates of gradients. Interestingly, recent work shows that despite the bias, given a large enough batch size, the direct approach produces almost optimal models (in terms of the marginal likelihood) (Chen et al., 2020).

Both the SVI approach and the direct approach suffer from several disadvantages. For example, consider the case in which the covariance function is the inner product between two feature maps of moderate dimension. This can be either because this decomposition serves as an approximation to another covariance function with an infinite dimensional *Reproducing Kernel Hilbert Space* (RKHS), or because it has been defined this way to begin with, e.g., as an inner product between features that are created by passing the data through a neural network. Either way, using the SVI approach amounts to imposing an additional unnecessary approximation that comes from the need to use inducing points. As for the second approach, its applicability is highly dependent on our ability to use large batches (Chen et al., 2016). This can be a serious impediment in several cases, such as optimization on weak edge devices.

In this paper, we propose two stochastic optimization algorithms based on mini-batches for maximizing the marginal likelihood of GPs (i.e., learning the hyperparameters). The first algorithm is based on reframing the problem as a nonconvex-concave minimax problem. We then leverage recent advancements in the theory of solving such problems (Boţ & Böhm, 2023; Lin et al., 2020; Luo et al., 2020) to propose a concrete algorithm. The second algorithm is based on writing the loss function in compositional form, i.e., as the composition of a function and the expected value of another function. We then use the recently introduced *Stochastic Compositional Gradient Descent* method (Wang et al., 2017). Our novel algorithms are applicable for covariance functions connected with an RKHS of moderate dimension and guarantee convergence of the marginal likelihood to a local minimum for any batch size without the need for further approximations. In the infinite dimensional case (e.g., Gaussian covariance function), one can use our method on top of a low rank approximation of the covariance function, e.g., using the *Random Fourier Features* method (Rahimi & Recht, 2007). Our experiments show that not only is our method superior to existing methods for stochastic optimization of the marginal likelihood in the finite dimensional case when the batches have a moderate size, but it is also superior to the existing methods in the infinite dimensional case if the restriction on batch size

is more severe. However, in the infinite dimensional case, the models found by our method are no longer optimal.

**Additional Related Work.** Apart from the extensive literature on scaling GPs using approximations, several works focus on exact inference using sophisticated distributed algorithms (Nguyen et al., 2019; Wang et al., 2019). In this context, the biased stochastic gradient proposed by Chen et al. (2020) can be considered an economical method for exact inference, given that the covariance function and the system enable computation in large enough batches.

## 2 Preliminaries

### 2.1 Notations and Basic Definitions

For a function $f:\mathcal{U} \to \mathbb{R}$ and $U = (u_1, \ldots, u_m) \in \mathcal{U}^m$, $\mathbf{f} = f(U)$ is a vector in $\mathbb{R}^m$ such that $\mathbf{f}_i = f(u_i)$. Similarly, for a binary function $k : \mathcal{U} \times \mathcal{U} \to \mathbb{R}$, $K = k(U, U)$ is a matrix in $\mathbb{R}^{m \times m}$ such that $K_{ij} = k(u_i, u_j)$. Given a size $m$ vector $\mathbf{b}$ and an index sequence $\mathcal{S} = (s_1, \ldots, s_r) \in \{1, \ldots, m\}^r$ (so $r = |\mathcal{S}|$), we use $\mathbf{b}_\mathcal{S}$ to denote the size $r$ vector such that the $i$th coordinate of $\mathbf{b}_\mathcal{S}$ is equal to the $s_i$th coordinate of $\mathbf{b}$. In a similar way, if $C$ is an $m \times n$ matrix, then $C_\mathcal{S}$ is a $r \times n$ matrix such that the $i$th row of $C_\mathcal{S}$ is equal to the $s_i$th row of $C$. Finally, if $\mathcal{R} = (r_1, \ldots, r_q) \in \{1, \ldots, n\}^q$, then $C_{\mathcal{S}\mathcal{R}}$ is a $r \times q$ matrix such that the $(i, j)$ coordinate of $C_{\mathcal{S}\mathcal{R}}$ is equal to the $(s_i, r_j)$ coordinate of $C$.

For a square matrix $A$ we use the notation $|A|$ to denote the determinant of $A$. If $\mathcal{S}$ is a finite sequence or a set then $|\mathcal{S}|$ denotes its length or size. Whenever we use $\langle A, B \rangle$, where $A$ and $B$ are real matrices, $\langle \cdot, \cdot \rangle$ symbolizes the Frobenius inner product, which is defined as

$$\langle A, B \rangle = \mathbf{Tr}\left(AB^T\right).$$

If $A$ is a real matrix, then $\|A\|$ is the Frobenius norm of $A$.

$$\|A\| = \sqrt{\langle A, A \rangle}.$$

For a vector $\mathbf{v}$, $\|\mathbf{v}\|$ is the Euclidean norm of $\mathbf{v} \in \mathbb{R}^q$.

For any closed convex set $\Omega \subseteq \mathbb{R}^q$ and $\mathbf{v} \in \mathbb{R}^q$, $\mathrm{proj}_\Omega(\mathbf{v}) = \arg\min_{\mathbf{v}' \in \Omega} \|\mathbf{v}' - \mathbf{v}\|$ is the Euclidean projection of $\mathbf{v}$ onto $\Omega$.

### 2.2 Hyperparameter Learning in Gaussian Process Regression

Let $\mathcal{X}$ be a feature space and let $k_\alpha : \mathcal{X} \times \mathcal{X} \to \mathbb{R}$ be a positive definite covariance function parameterized by hyperparameters $\alpha \in \mathbb{R}^m$. For any $\alpha \in \mathbb{R}^m$, let $f_\alpha$ be a random function on $\mathcal{X}$ distributed as a zero mean GP whose covariance is $k_\alpha$, that is, for any $j \in \mathbb{N}$, $U \in \mathcal{X}^j$:

$$f_\alpha(U) \sim \mathcal{N}(0, k_\alpha(U, U)).$$

Let $X \in \mathcal{X}^n$, $\mathbf{y} \in \mathbb{R}^n$ be a training set, where $n$ is the number of training samples. In *Gaussian Process Regression* (GPR), it is assumed that $\mathbf{y}$ is a sample of the random vector $f_{\bar\alpha}(X) + \epsilon$, where $\epsilon \sim \mathcal{N}\left(0, \bar\sigma^2 I_n\right)$ and $\bar\alpha, \bar\sigma^2$ are the true hyperparameters of the model.

In GPR, hyperparameters are learned by solving a maximum likelihood type II problem, i.e., maximizing the marginal likelihood

$$p\left(\mathbf{y}|X, \alpha, \sigma^2\right) = \mathcal{N}\left(\mathbf{y}|0, K(\alpha) + \sigma^2 I\right),$$

with respect to $\alpha$ and $\sigma^2 > \sigma^2_{\min} > 0$, where $K(\alpha) = k_\alpha(X, X)$. The maximization of the marginal likelihood $p\left(\mathbf{y}|X, \alpha, \sigma^2\right)$ is equivalent to the minimization of

$$\mathbf{y}^T\left(K(\alpha) + \sigma^2 I\right)^{-1}\mathbf{y} + \log\left|K(\alpha) + \sigma^2 I\right|.$$

See Rasmussen (2003) for details. In this work, we further assume that the covariance function has the following form:

$$k_\alpha (x, x') = \phi_\alpha (x)^T \phi_\alpha (x'),$$

for some feature map $\phi_\alpha : \mathcal{X} \to \mathbb{R}^d$.

The feature map $\phi_\alpha$ may also arise from a fixed finite-rank approximation of an infinite-dimensional kernel. For example, if a Nyström landmark set is selected before optimization and then kept fixed, the resulting Nyström features define a finite-dimensional map of the above form (Williams & Seeger, 2001; Quinonero-Candela & Rasmussen, 2005). Our algorithms then optimize the marginal likelihood of the corresponding finite-feature approximate GP.

It can be shown that for all $\lambda > 0$, $V \in \mathbb{R}^{n \times d}$, and $\mathbf{b} \in \mathbb{R}^n$, we have that

$$\mathbf{b}^T \left( VV^T + \lambda I \right)^{-1} \mathbf{b} = \min_{\mathbf{w}} \frac{1}{\lambda} \|V\mathbf{w} - \mathbf{b}\|^2 + \|\mathbf{w}\|^2,$$

(see Appendix B). As a result, maximizing $p(\mathbf{y}|X, \alpha, \sigma)$ is equivalent to minimizing

$$\ell(\theta) = \frac{1}{\sigma^2} \|Z(\alpha) \mathbf{w} - \mathbf{y}\|^2 + \|\mathbf{w}\|^2 + \log |F(\theta)| + (n - d) \log \sigma^2,$$

where

$$\theta = (\mathbf{w}, \alpha, \beta),$$
$$\beta = \log \left( \sigma^2 - \sigma_{\min}^2 \right),$$
$$Z(\alpha) = (\phi_\alpha (x_1), \dots, \phi_\alpha (x_n))^T \in \mathbb{R}^{n \times d},$$
$$F(\theta) = Z(\alpha)^T Z(\alpha) + \sigma^2 I \in \mathbb{R}^{d \times d}.$$

The goal of this work is to propose algorithms for minimizing $\ell(\theta)$ using a stochastic gradient method based on mini-batches. This would have been straightforward if we could write

$$\nabla_\theta \log |F(\theta)| = \sum_{i=1}^n G(\theta; x_i),$$

for some function $G(\cdot; \cdot)$. However, there is no obvious decomposition of this form.

## 3 Stochastic Optimization for Gaussian Processes

This section contains our main contribution: two novel stochastic mini-batch-based algorithms for minimizing $\ell(\theta)$. Before detailing our approaches, we make a few additional notations:

$$g_i(\theta) = \frac{1}{\sigma^2} \left( \phi_\alpha (x_i)^T \mathbf{w} - y_i \right)^2 + \frac{1}{n} \|\mathbf{w}\|^2 + \frac{1}{n} (n - d) \log \sigma^2,$$
$$g(\theta) = \sum_{i=1}^n g_i(\theta),$$
$$F_i(\theta) = \phi_\alpha (x_i) \phi_\alpha (x_i)^T + \frac{1}{n} \sigma^2 I,$$
$$h(A) = \log |A|.$$

So, we can write,

$$F(\theta) = \sum_{i=1}^n F_i(\theta),$$
$$\ell(\theta) = g(\theta) + h(F(\theta)).$$

### 3.1 A Minimax Approach

Our first step is to replace the problem

$$\min_\theta g\left(\theta\right) + h\left(F\left(\theta\right)\right),$$

with the equivalent problem

$$\min_{\theta,A} g\left(\theta\right) + h\left(A\right)$$
$$\text{s.t. } A = F\left(\theta\right). \tag{1}$$

The next step is to replace the last problem with a parameterized problem, such that the hard constraint is replaced with a penalty term, and the penalty term is driven to infinity. To do so, let us first define the optimization problems

$$\min_\zeta \ell_\mu\left(\zeta\right)$$

where $\zeta = \left(\theta, A\right)$ and

$$\ell_\mu\left(\zeta\right) = g\left(\theta\right) + h\left(A\right) + \mu\frac{\|A - F\left(\theta\right)\|}{\|A\|}.$$

Now, suppose that Problem (1) admits an optimal solution. Let the sequences $\{\mu_k\}$ and $\{\zeta_k\}$ satisfy $\mu_k \to \infty$ and, for each $k$, let $\zeta_k$ minimize $\ell_{\mu_k}(\zeta)$. If $\zeta^\star$ is an accumulation point of $\{\zeta_k\}$, then $\zeta^\star$ is an optimal solution of Problem (1) (Ruszczyński, Theorem 6.6).

We remark that, rather than minimizing only $\|A - F(\theta)\|$, we use the modified objective described above. Empirically, this reduces convergence to undesirable stationary points, e.g., runs in which $F(\theta)$ becomes spuriously small without improving downstream performance.

Unfortunately, the term $\|A - F\left(\theta\right)\|$ exhibits the same issues as $h\left(F\left(\theta\right)\right) = \log|F(\theta)|$ and does not allow for straightforward unbiased mini-batch-based stochastic gradients. In order to handle this, we use the fact that for an element $v$ in Euclidean space, we have that $\|v\| = \max_{\|u\|\leq 1} \langle u, v\rangle$. Thus, minimizing $\ell_\mu\left(\zeta\right)$ is equivalent to

$$\min_\zeta \max_{\|B\|\leq 1} \Psi\left(\zeta, B; \mu\right), \tag{2}$$

where

$$\Psi\left(\zeta, B; \mu\right) = g\left(\theta\right) + h\left(A\right) + \mu\frac{\langle B, A - F\left(\theta\right)\rangle}{\|A\|},$$

Now, we can write

$$\Psi\left(\zeta, B; \mu\right) = \sum_{i=1}^n \psi\left(\zeta, B; x_i, \mu\right), \tag{3}$$

where

$$\psi\left(\zeta, B; x_i, \mu\right) = g_i\left(\theta\right) + \frac{1}{n}h\left(A\right) + \mu\frac{\langle B, \frac{1}{n}A - F_i\left(\theta\right)\rangle}{\|A\|},$$

Thus, Problem (2) naturally admits unbiased stochastic gradients based on mini-batches taken separately for the minimization and maximization parts.

For numerical stability, we keep the (learned) noise variance bounded away from zero and ensure $A$ stays uniformly well-conditioned by enforcing

$$\sigma^2 > \sigma_{\min}^2 > 0 \quad, \sigma^2 I \preceq A \preceq A_{\max}.$$

In addition, we clip the model parameters $\theta = (\mathbf{w}, \alpha, \beta) = (\mathbf{w}, \alpha, \log(\sigma^2 - \sigma_{\min}^2))$ to a box

$$\theta_{\min} \preceq \theta \preceq \theta_{\max}.$$

Altogether, the feasible set for $\zeta = (\theta, A)$ is

$$\Omega_1 = \Big\{ (\theta, A) \ \Big| \ \theta_{\min} \preceq \theta \preceq \theta_{\max}, \ \sigma^2 I \preceq A \preceq A_{\max} \Big\},$$

where $\sigma_{\min}^2 > 0$, $\sigma_{\max}^2$, $\theta_{\min}$, $\theta_{\max}$, and $A_{\max}$ are fixed (non-learned) hyperparameters.

Let $\Omega_2 = \big\{ B \in \mathbb{R}^{d \times d} \mid \|B\| \leq 1 \big\}$. The algorithm we propose is based on the following update rule for solving Problem (2):

1. $\zeta_{t+1} = \mathrm{proj}_{\Omega_1} \left( \zeta_t - a \nabla_\zeta \dfrac{n}{s} \displaystyle\sum_{i \in \mathcal{S}_{t+1}} \psi\left(\zeta_t, B_t; x_i, \mu\right) \right),$

2. $B_{t+1} = \mathrm{proj}_{\Omega_2} \left( B_t + b \nabla_B \dfrac{n}{s} \displaystyle\sum_{i \in \bar{\mathcal{S}}_{t+1}} \psi\left(\zeta_{t+1}, B_t; x_i, \mu\right) \right).$

$$\mu_{k+1} = \mathcal{U}(\mu_k), \qquad \mu_k \uparrow \infty.$$

where $\mathcal{S}_t$ and $\bar{\mathcal{S}}_t$ are sets of $s$ indices randomly chosen from $\{1, \ldots, n\}$, independently from all previous iterations. Recent work by Boţ & Böhm (2023) shows that in the case of a minimax problem where the maximization is of a concave function over a convex constraint, for correctly chosen step sizes $a$ and $b$, [1] with a few additional mild conditions on the objective function, an algorithm based on the above update rule visits a point that is at an $O(\epsilon)$ distance from a stationary point in $O\left(\epsilon^{-8}\right)$ iterations (more details can be found in Appendix B.2).

While the projection onto $\Omega_2$ is just the Euclidean projection onto the Frobenius unit ball, the projection onto $\Omega_1$ consists of (i) simple coordinate-wise clipping of the vector parameters in $\theta$ and (ii) spectral clipping of $A$ to satisfy the matrix inequalities $\sigma^2 I_d \preceq A \preceq A_{\max}$. Concretely, letting $A = UDU^\top$ be an eigendecomposition and assuming $A_{\max}$ is diagonal (e.g., $A_{\max} = a_{\max} I_d$), we compute

$$\mathrm{proj}_{\left\{ \sigma^2 I_d \preceq M \preceq A_{\max} \right\}}(A) = U \, \mathrm{diag}\Big( \min\big(\max(D_{11}, \sigma^2), (A_{\max})_{11}\big), \ldots, \min\big(\max(D_{dd}, \sigma^2), (A_{\max})_{dd}\big) \Big) U^\top.$$

This projection therefore requires an eigenvalue decomposition of a $d \times d$ matrix, but it does not change the overall per-iteration asymptotic complexity; see Appendix A for details.

The Minimax approach is summarized in Algorithm 1.

## 3.2 Stochastic Compositional Gradient Descent Approach

Consider a loss function $\ell : \Theta \to \mathbb{R}$ of the form $\ell(\theta) = v(u(\theta))$ where $u : \Theta \to \mathbb{R}^p$ and $v : \mathbb{R}^p \to \mathbb{R}$ are differentiable functions, and assume that $u(\theta) = \mathbb{E}_\omega \tilde{u}(\theta; \omega)$ for a differentiable function $\tilde{u}(\theta; \omega)$ that depends on a random variable $\omega$. *Stochastic Compositional Gradient Descent* (SCGD) (Wang et al., 2017) is an intuitive algorithm that alternates between two steps: updating the solution $\theta_t$ by a stochastic gradient iteration, and estimating $u(\theta_t)$ using an iterative weighted average of past values. More precisely, the update rule of SCGD is given by:

$$\eta_{t+1} = (1 - b_t) \eta_t + b_t \, \tilde{u}(\theta_t; \omega_t),$$
$$\theta_{t+1} = \theta_t - a_t \, J_t^\top \, \nabla v(\eta_{t+1}).$$

where $\omega_1, \omega_2, \ldots$ are samples from $\omega$ in an i.i.d. manner, and $a_0, a_1, \ldots, b_0, b_1, \ldots$ degrees of freedom in the algorithm. Under a few additional standard conditions on $u, v, \tilde{u}$, Wang et al. (2017) establish an iteration–complexity bound for basic SCGD in the sense that, if we define

$$T_\epsilon := \min\Big\{ t \geq 0 : \inf_{0 \leq s \leq t} \mathbb{E}\big[\|\nabla \ell(\theta_s)\|^2\big] \leq \epsilon \Big\},$$

---
[1]The setup in Boţ & Böhm (2023) is more general.

---

**Algorithm 1** Stochastic Minimax (Alternating) for GP Hyperparameter Learning (Outer-Loop Penalty)

---

**Require:** Dataset $(X, \mathbf{y})$ of size $n$, feature map $\phi_\alpha$, batch size $s$
**Require:** Inner iterations $T$, outer penalty stages $K$
**Require:** Step sizes $a, b > 0$
**Require:** Penalty initialization $\mu_0 > 0$ and a nondecreasing update rule $\mu_{k+1} = \mathcal{U}(\mu_k)$
**Require: (Fixed hyperparameters defining the constraint sets)** box bounds $\theta_{\min}, \theta_{\max}$ and a PSD upper bound $A_{\max} \succeq 0$.

1: **Definitions:** ($\preceq$ is elementwise for vectors and Loewner order for matrices; $I_d$ is the $d \times d$ identity)

$$\Omega_1 = \left\{ \zeta = (\theta, A) : \; \theta_{\min} \preceq \theta \preceq \theta_{\max}, \; \sigma^2 I_d \preceq A \preceq A_{\max} \right\}$$

$$\Omega_2 = \{ B \in \mathbb{R}^{d \times d} : \|B\| \leq 1 \}$$

$$g_i(\theta) = \frac{1}{\sigma^2} \left( \phi_\alpha(x_i)^\top \mathbf{w} - y_i \right)^2 + \frac{1}{n} \|\mathbf{w}\|^2 + \frac{1}{n}(n - d) \log \sigma^2$$

$$F_i(\theta) = \phi_\alpha(x_i) \phi_\alpha(x_i)^\top + \frac{1}{n} \sigma^2 I_d, \quad h(A) = \log |A|$$

$$\psi(\zeta, B; x_i, \mu) = g_i(\theta) + \frac{1}{n} h(A) + \mu \frac{\langle B, \; \frac{1}{n} A - F_i(\theta) \rangle}{\|A\|}$$

2: Initialize $\zeta \leftarrow (\theta_0, A_0) \in \Omega_1$, $B \leftarrow B_0 \in \Omega_2$, and $\mu \leftarrow \mu_0$
3: **for** $k = 0$ to $K - 1$ **do** $\qquad\qquad\qquad\qquad\qquad\qquad\qquad\qquad\qquad$ $\triangleright$ outer penalty loop
4: $\quad$ **for** $t = 0$ to $T - 1$ **do** $\qquad\qquad\qquad\qquad\qquad\qquad\qquad$ $\triangleright$ inner minimax loop at fixed $\mu$
5: $\qquad$ Sample mini-batches $\mathcal{S}, \bar{\mathcal{S}} \subset \{1, \ldots, n\}$ with $|\mathcal{S}| = |\bar{\mathcal{S}}| = s$ independently
6: $\qquad$ **(Min step)** Compute

$$G_\zeta \leftarrow \frac{n}{s} \sum_{i \in \mathcal{S}} \nabla_\zeta \psi(\zeta, B; x_i, \mu)$$

7: $\qquad\qquad$ Update $\zeta \; \leftarrow \; \text{proj}_{\Omega_1} \left( \zeta - a \, G_\zeta \right)$
8: $\qquad$ **(Max step)** Compute

$$G_B \leftarrow \frac{n}{s} \sum_{i \in \bar{\mathcal{S}}} \nabla_B \psi(\zeta, B; x_i, \mu)$$

9: $\qquad\qquad$ Update $B \; \leftarrow \; \text{proj}_{\Omega_2} \left( B + b \, G_B \right)$
10: $\quad$ **end for**
11: $\quad$ **(Penalty update)** $\mu \; \leftarrow \; \mathcal{U}(\mu)$ $\qquad\qquad$ $\triangleright$ e.g., $\mu \leftarrow \gamma\mu$ with $\gamma \geq 1$; constant-$\mu$ uses $\gamma = 1$
12: **end for**
13: **Return** $\theta = (\mathbf{w}, \alpha, \beta)$ from $\zeta$

---

then, for power-law step sizes $a_t = (t + 1)^{-\gamma}$ and $b_t = (t + 1)^{-\delta}$ with $\delta < \gamma < 2\delta$, we have

$$T_\epsilon = \mathcal{O}\left( \epsilon^{-1/q} \right), \qquad q = \min\{ 1 - \gamma, \; \gamma - \delta, \; 2\delta - \gamma, \; \gamma \}.$$

In particular, taking $(\gamma, \delta) = \left( \frac{3}{4}, \frac{1}{2} \right)$ yields $q = \frac{1}{4}$ and hence recovers $T_\epsilon = \mathcal{O}\left( \epsilon^{-4} \right)$ (see Wang et al. (2017, Theorem 4) and Appendix B, Theorem 3).

In order to use SCGD for our loss, we define:

$$u(\theta) = (g(\theta), F(\theta)),$$

$$v(u_1, u_2) = u_1 + h(u_2),$$

$$\tilde{u}(\theta; \omega) = \frac{n}{|\mathcal{S}|} \sum_{i \in \mathcal{S}} (g_i(\theta), F_i(\theta)),$$

where $\omega = \mathcal{S}$ is a random set of indices chosen from $\{1, \ldots, n\}$. With that, given the fact that

$$\frac{\partial \log |A|}{\partial A} = A^{-1}$$

---

**Algorithm 2** SCGD for GP Hyperparameter Learning

---

**Require:** Dataset $(X, \mathbf{y})$ of size $n$, feature map $\phi_\alpha$, batch size $s$, iterations $T$
**Require:** Step-size sequences $\{a_t\}_{t=0}^{T-1}$ and $\{b_t\}_{t=0}^{T-1}$ (e.g., $a_t = (t+1)^{-3/4}$, $b_t = (t+1)^{-1/2}$)
1: **Notation:** $d = \dim\phi_\alpha(\cdot)$, $I_d$ is the $d \times d$ identity ($\preceq$ is elementwise for vectors)
2: **Per-sample terms:**

$$g_i(\theta) = \frac{1}{\sigma^2}\big(\phi_\alpha(x_i)^\top \mathbf{w} - y_i\big)^2 + \frac{1}{n}\|\mathbf{w}\|^2 + \frac{1}{n}(n-d)\log\sigma^2$$

$$F_i(\theta) = \phi_\alpha(x_i)\phi_\alpha(x_i)^\top + \frac{1}{n}\sigma^2 I_d$$

3: Initialize $\theta_0 \in \Theta$ and a SPD tracker $\tilde{F}_0 \succ 0$ (e.g., $\tilde{F}_0 \leftarrow \sigma_0^2 I_d$)
4: **for** $t = 0$ **to** $T-1$ **do**
5:     Sample mini-batch $\mathcal{S}_{t+1} \subset \{1, \ldots, n\}$, $|\mathcal{S}_{t+1}| = s$
6:     **(Exponential tracking of $F(\theta)$)**

$$\tilde{F}_{t+1} \;\leftarrow\; (1 - b_t)\,\tilde{F}_t \;+\; b_t\,\frac{n}{s}\sum_{i \in \mathcal{S}_{t+1}} F_i(\theta_t)$$

7:     Compute $\tilde{F}_{t+1}^{-1}$ (e.g., via Cholesky of $\tilde{F}_{t+1}$)
8:     **(Gradient step)**

$$G_\theta \leftarrow \frac{n}{s}\sum_{i \in \mathcal{S}_{t+1}} \nabla_\theta\Big[g_i(\theta) + \langle \tilde{F}_{t+1}^{-1}, F_i(\theta)\rangle\Big]\Big|_{\theta = \theta_t}$$

9:     $\theta_{t+1} \;\leftarrow\; \theta_t - a_t\, G_\theta$
10: **end for**
11: **Return** $\theta_T = (\mathbf{w}_T, \alpha_T, \beta_T)$

---

we can write an explicit update rule for our SCGD-based algorithm:

$$\tilde{F}_{t+1} = (1 - b_t)\,\tilde{F}_t \;+\; b_t\,\frac{n}{|\mathcal{S}_{t+1}|}\sum_{i \in \mathcal{S}_{t+1}} F_i(\theta_t)\,,$$

$$\theta_{t+1} = \theta_t \;-\; a_t\,\frac{n}{|\mathcal{S}_{t+1}|}\sum_{i \in \mathcal{S}_{t+1}} \nabla_\theta\big[g_i(\theta) + \big\langle \tilde{F}_{t+1}^{-1}, F_i(\theta)\big\rangle\big]\big|_{\theta = \theta_t}\,.$$

The SCGD approach is summarized in Algorithm 2. A formal convergence theorem can be found in Appendix B, Theorem 3.

## 4 Comparison to Existing Methods

In this work, we suggest two novel methods for the stochastic optimization of the marginal likelihood. We recognize two main competing methods that also optimize the marginal likelihood stochastically. The simplest among them is what we henceforth refer to as *Biased Stochastic Gradient Descent* (BSGD) (Chen et al., 2020). The idea in BSGD is to take the gradient of the marginal likelihood using data only from the current batch, ignoring the fact that this produces only a biased estimate of the full gradient. On the other hand, *the Scalable Variational Gaussian Process* (SVGP) of Hensman et al. (2015) is a sophisticated approach that approximates the original inference problem using inducing points and stochastic variational inference.

Unlike our algorithms, the effectiveness of each of the competing methods in approximating the solution to the true problem depends on memory consumption. In the case of BSGD, the bias in stochastic gradients shrinks as the batch size increases. On the other hand, the inference quality of SVGP crucially depends on the number of inducing points that need to be processed forward and backward by a neural network at each iteration.

Table 1: Complexity analysis for one optimization iteration. Here $b$ is the data mini-batch size, $m$ is the number of SVGP inducing points, $d$ is the feature dimension, and C and M are the per-point computational and memory costs of the feature map and its derivative.

| Algorithm | Computations | Storage |
|---|---|---|
| SVGP | $O(m^3) + O(bm^2) + O((bm + m^2)d) + (b+m)\text{C}$ | $O(m^2 + bm) + O((b+m)d) + (b+m)\text{M}$ |
| BSGD | $O(b^3) + O(b^2 d) + b\text{C}$ | $O(b^2) + O(bd) + b\text{M}$ |
| Minimax | $O(d^3) + O(bd^2) + b\text{C}$ | $O(d^2) + O(bd) + b\text{M}$ |
| SCGD | $O(d^3) + O(bd^2) + b\text{C}$ | $O(d^2) + O(bd) + b\text{M}$ |

## 4.1 Complexity Analysis

Let $b$ be the data mini-batch size, $m$ the number of inducing points used by SVGP, and $d$ the feature dimension. Let C and M be, respectively, the per-point computational and memory costs of evaluating $\phi_\alpha(x)$ and differentiating it with respect to $\alpha$. BSGD and our algorithms process $b$ points through the feature map per iteration, whereas SVGP processes $b + m$ points.

Each inducing point must be processed by the same feature map (i.e., it requires a forward and backward pass through the network), so it incurs essentially the same activation and gradient-storage cost as an additional training example. Consequently, setting $m = b$, as in our experiments, gives SVGP approximately twice as many feature-map passes per iteration as the other methods.

The resulting per-iteration computational costs are $O(b^3) + O(b^2 d) + b\text{C}$ for BSGD, $O(m^3) + O(bm^2) + O((bm + m^2)d) + (b+m)\text{C}$ for SVGP, and $O(d^3) + O(bd^2) + b\text{C}$ for each of our algorithms. The corresponding storage costs are $O(b^2) + O(bd) + b\text{M}$ for BSGD, $O(m^2 + bm) + O((b+m)d) + (b+m)\text{M}$ for SVGP, and $O(d^2) + O(bd) + b\text{M}$ for our algorithms.

Typically, M is relatively large, as it is the storage used by *Automatic Differentiation* (AD) to compute the gradient of a neural network. In small devices, this means one might have to use small batches. Our approach aims not to harm the exactness of the algorithm in this case. Naturally, to use our algorithms effectively, the feature dimension $d = \dim \phi_\alpha(x)$ must remain moderate, since memory scales as $O(d^2)$ and computation as $O(d^3)$.

## 4.2 Relation to BSGD

Our approach is close to the BSGD approach of Chen et al. (2020). BSGD is based on the following update formula:

$$\theta_{t+1} = \theta_t - a_t \nabla l(\theta_t; \mathcal{S}_t),$$

where $\mathcal{S}_t$ are an independent random batch of indices and

$$\ell(\theta; \mathcal{S}) = \mathbf{y}_\mathcal{S}^T \left( K_{\mathcal{S}\mathcal{S}}(\theta) + \sigma^2 I \right)^{-1} \mathbf{y}_\mathcal{S} + \log \left| K_{\mathcal{S}\mathcal{S}}(\theta) + \sigma^2 I \right|.$$

In our case, since $k_\alpha(x) = \phi_\alpha(x)^T \phi_\alpha(x)$, we can write $l(\theta; \mathcal{S})$ as

$$\ell(\theta; \mathcal{S}) = \sum_i g_i(\theta) + h\left( \sum_i F_i(\theta) \right).$$

This leads to the update formula

$$\theta_{t+1} = \theta_t - a_t \sum_{i \in \mathcal{S}} \nabla \left[ g_i(\theta_t) + \left\langle \tilde{F}_t^{-1}, F_i(\theta) \right\rangle \right], \text{ where } \tilde{F}_t = \sum_{i \in \mathcal{S}} F_i(\theta_t).$$

Writing the update formula of the BSGD algorithm this way emphasizes the fact that the only difference between the BSGD and the SCGD algorithms lays in the way in which $F(\theta_t)$ is approximated: using $\tilde{F}_t = \sum_{i \in \mathcal{S}} F_i(\theta_t)$ in BSGD, and $\tilde{F}_{t+1} = (1 - b_t)\tilde{F}_t + b_t \frac{n}{|\mathcal{S}|} \sum_{i \in \mathcal{S}} F_i(\theta_t)$ in SCGD. Intuitively, the exponential smoothing that occurs SCGD should provide additional numerical robustness beyond the theoretical advantage of converging to a stationary point without an error which depends on the batch size. We can also see that in BSGD the scale of $\tilde{F}_t$ is incorrect because it misses the multiplication of $\sum_{i \in \mathcal{S}} F_i(\theta_t)$ by $\frac{n}{|\mathcal{S}|}$.

## 5 Experimental Result

### 5.1 Recovery of Known Parameters

In this set of experiments, we follow the synthetic experiment of Chen et al. (2020). We sample $n = 1024$ one-dimensional inputs $x_i \overset{i.i.d.}{\sim} \mathcal{N}(0, 5^2)$ and generate responses from a zero-mean GP with two variance parameters,

$$\mathbf{y} \sim \mathcal{N}(0, \ \sigma_f^2 K_f + \sigma_\varepsilon^2 I_n),$$

where the lengthscale is fixed to $\ell = 0.5$ and the ground-truth values are $\sigma_f^2 = 4$ and $\sigma_\varepsilon^2 = 1$.

In order to stay in the exact finite dimensional setting of this paper, we construct $K_f$ via a fixed feature map $\varphi : \mathbb{R} \to \mathbb{R}^d$ with $d = 128$ such that

$$k_f(x, x') = \langle \varphi(x), \varphi(x') \rangle, \qquad K_f = k_f(X, X) = \varphi(X)\varphi(X)^T.$$

Concretely, we obtain $\varphi$ using *Orthogonal Random Features* (ORF) (Yu et al., 2016) with the same lengthscale $\ell$. With this, the learnable kernel magnitude can be written in the form assumed throughout the paper,

$$k_\alpha(x, x') = \phi_\alpha(x)^T \phi_\alpha(x'), \qquad \phi_\alpha(x) = \sqrt{\sigma_f^2} \ \varphi(x),$$

so here $\alpha = \sigma_f^2$ is the only kernel hyperparameter we learn, and the observation-noise variance corresponds to $\sigma^2 = \sigma_\varepsilon^2$ in our notation. We optimize $\theta = (\mathbf{w}, \sigma_f^2, \sigma_\varepsilon^2)$ using Algorithm 2 with mini-batches of size $s = 128$.

We used step-size schedules of the form $a_t = a_0 \, t^{-0.3}$ and $b_t = b_0 \, t^{-0.2}$ with $a_0 = 10^{-3}$ and $b_0 = 0.1$. We ran SCGD for $2 \times 10^6$ epochs and, for readability, plot the first $5 \times 10^4$ epochs (i.e., $4 \times 10^5$ stochastic updates). We repeated the experiment over 10 independent random seeds and, as in Chen et al. (2020), we considered three different initializations for the variance parameters: $(\sigma_{f,0}^2, \sigma_{\varepsilon,0}^2) \in \{(5.0, 3.0), (2.5, 3.5), (2.5, 0.7)\}$.

Figure 1 shows that SCGD converges to a neighborhood of the true parameters from all three initial points. As in Chen et al. (2020), the trajectories of $\sigma_\varepsilon^2$ are markedly more concentrated around the truth than those of $\sigma_f^2$, reflecting the fact that the signal-variance estimate exhibits a larger statistical variability. Most importantly for our purposes, this experiment is carried out in the exact inner-product setting $K_f = \varphi(X)\varphi(X)^T$, and therefore the observed behavior reflects optimization of the true objective by unbiased mini-batch gradients (rather than an artifact of submatrix bias).

**MINIMAX.** We also repeat the above experiment using our MINIMAX procedure (Algorithm 1). Although the penalized formulation recovers the original objective as $\mu \to \infty$, increasing $\mu$ is well known to make penalty methods progressively ill-conditioned in practice (Nocedal & Wright, 2006). We therefore treat $\mu$ as a tunable constant and study its effect on how closely MINIMAX recovers the same stationary point as full-batch optimization of the exact objective.

We use the same dataset $(X, \mathbf{y})$ and the same ORF feature map $\varphi$ (and therefore the same finite dimensional kernel $K_f = \varphi(X)\varphi(X)^\top$), and we fix the mini-batch size to $s = 128$. We initialize the hyperparameters at $(\sigma_f^2, \sigma_\varepsilon^2) = (3.0, 2.0)$ and run MINIMAX for 40,000 iterations with constant step sizes $a = 10^{-5}$ and $b = 4 \times 10^{-4}$. We sweep the penalty parameter over $\mu \in \{0, 1, 10, 10^2, 10^3\}$ and repeat each setting over three random seeds.

For each seed, we also compute a reference solution by running full-batch gradient descent on the *exact* negative log marginal likelihood until convergence, and denote the resulting stationary point by $(\sigma_f^2)^\star$ and $(\sigma_\varepsilon^2)^\star$.

Figure 2 shows a clear monotone effect of the penalty parameter: larger values of $\mu$ yield iterates whose variance estimates and attained negative log marginal likelihood track the full-batch reference increasingly closely. Overall, this sweep illustrates the practical role of $\mu$ in MINIMAX: increasing $\mu$ improves fidelity to the exact objective. However, taking $\mu$ too large can lead to numerical instability; while this is not observed in Figure 2, in our experiments, such issues begin to appear for $\mu$ somewhere above $10^3$.

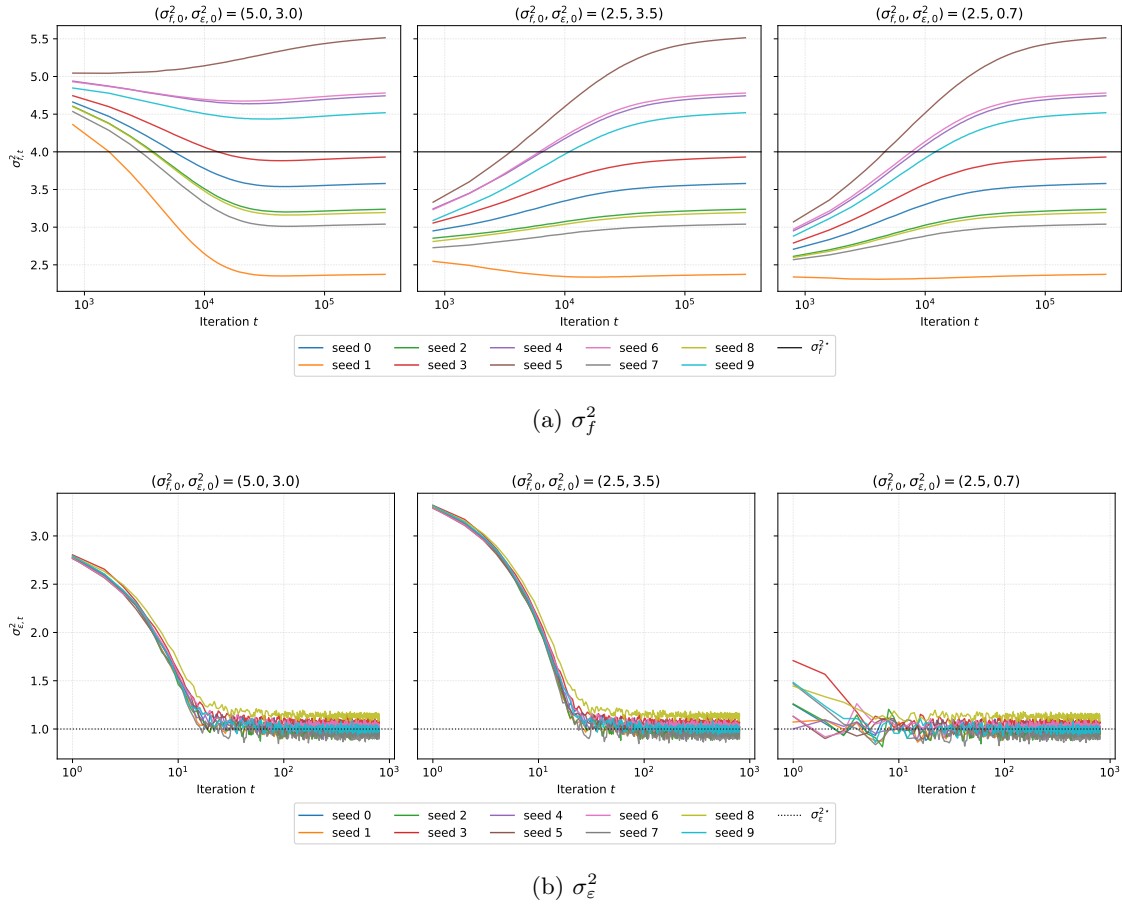

(a) $\sigma_f^2$

(b) $\sigma_\varepsilon^2$

Figure 1: Recovery of known variance parameters with SCGD (finite dimensional setting, SCGD+ORF). Each color corresponds to a different random seed. The horizontal black lines denote the ground-truth values $\sigma_f^2 = 4$ (top) and $\sigma_\varepsilon^2 = 1$ (bottom).

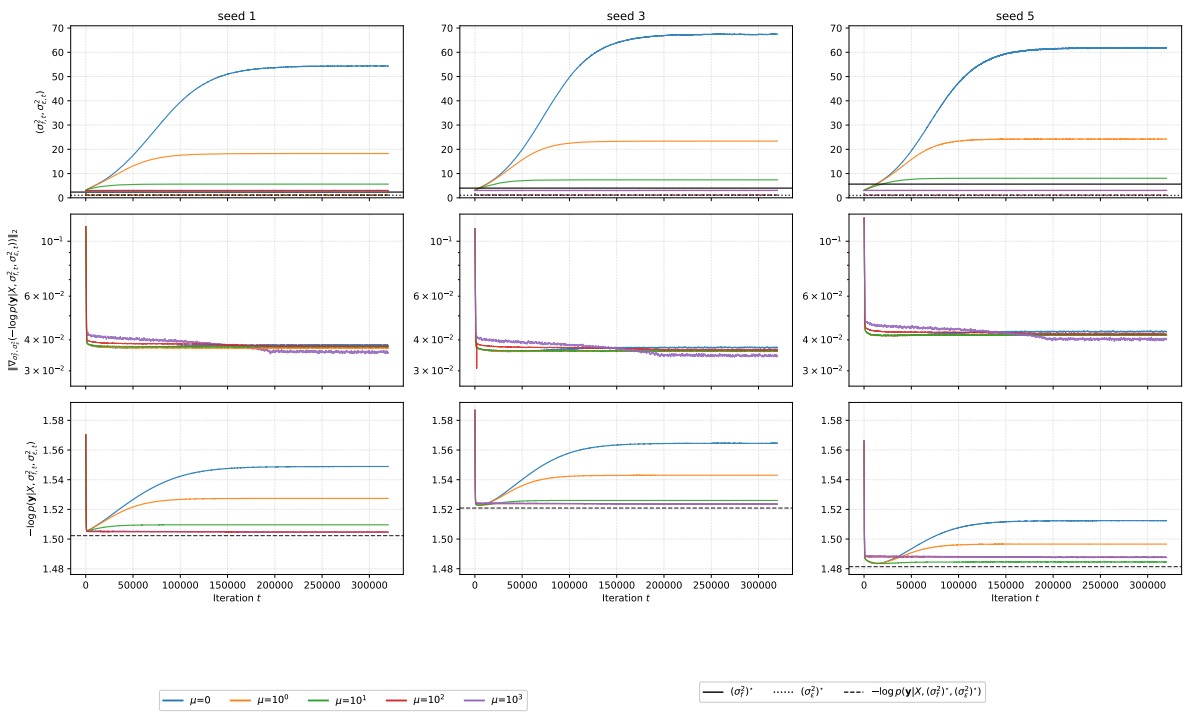

Figure 2: Effect of the penalty weight $\mu$ for MINIMAX on the synthetic recovery experiment. Each column corresponds to a different random seed. Top: MINIMAX estimates of $\sigma_f^2$ (solid) and $\sigma_\varepsilon^2$ (dashed) as a function of iteration, for several fixed choices of $\mu$. Middle: exact full-batch gradient norm. Bottom: exact per-sample negative log marginal likelihood.

Table 2: Win/loss counts comparing our algorithms with the existing methods for each batch size. Each row counts outcomes across the 9 UCI datasets.

(a) Linear kernel, negative log marginal likelihood

| | MINIMAX | | SCGD | |
| --- | --- | --- | --- | --- |
| Batch size | Wins | Losses | Wins | Losses |
| 32 | 7 | 2 | 7 | 2 |
| 64 | 7 | 2 | 7 | 2 |
| 128 | 6 | 3 | 7 | 2 |
| 256 | 5 | 4 | 3 | 6 |
| 512 | 5 | 4 | 5 | 4 |

(b) Linear kernel, test RMSE

| | MINIMAX | | SCGD | |
| --- | --- | --- | --- | --- |
| Batch size | Wins | Losses | Wins | Losses |
| 32 | 4 | 5 | 3 | 6 |
| 64 | 5 | 4 | 4 | 5 |
| 128 | 5 | 4 | 4 | 5 |
| 256 | 2 | 7 | 1 | 8 |
| 512 | 2 | 7 | 1 | 8 |

(c) Gaussian kernel, negative log marginal likelihood

| | MINIMAX | | SCGD | |
| --- | --- | --- | --- | --- |
| Batch size | Wins | Losses | Wins | Losses |
| 16 | 5 | 4 | 6 | 3 |
| 32 | 5 | 4 | 6 | 3 |
| 64 | 4 | 5 | 5 | 4 |
| 128 | 4 | 5 | 5 | 4 |
| 256 | 1 | 8 | 4 | 5 |

(d) Gaussian kernel, test RMSE

| | MINIMAX | | SCGD | |
| --- | --- | --- | --- | --- |
| Batch size | Wins | Losses | Wins | Losses |
| 16 | 5 | 4 | 5 | 4 |
| 32 | 5 | 4 | 5 | 4 |
| 64 | 5 | 4 | 4 | 5 |
| 128 | 2 | 7 | 2 | 7 |
| 256 | 3 | 6 | 3 | 6 |

### 5.2 GP Regression

In this set of experiments, we consider a covariance function of the form

$$k_\alpha\left(x, x'\right) = k'_u\left(g_w\left(x\right), g_w\left(x'\right)\right),$$

where $g_w$ is a neural network comprising two fully connected layers, both with an output dimension of 128 and a ReLU activation function , and $k'_u$, which is either the linear kernel

$$k'_u\left(z, z'\right) = \langle z, z'\rangle,$$

or the Gaussian kernel parametrized with two hyperparameters, length scale $u_1$ and magnitude $u_2$, that is

$$k'_u\left(z, z'\right) = u_2 e^{-\frac{\|z-z'\|^2}{2u_1^2}}.$$

Note that the accuracy of both competing methods depends on memory consumption. In the case of BSGD, the stochastic gradient becomes more unbiased as the batch size increases. On the other hand, the quality of the inference of the SVGP algorithm depends on the number of inducing points that need to be processed, both forward and backward, by the neural network at each iteration.

**Setup:** The experiment is designed to observe the influence of batch size on the results. We used 9 regression datasets from the UCI repository (Asuncion & Newman, 2007), all with a number of samples above 14,000 and fewer than 60,000. We tested the algorithms using different batch sizes: for the linear kernel, we examined batch sizes of 32, 64, 128, 256, and 512; for the Gaussian kernel, we examined batch sizes of 16, 32, 64, 128, and 256. For SVGP, we set the number of inducing points equal to the mini-batch size, as a simple balanced allocation. For a fixed data mini-batch size, this gives SVGP roughly twice as many forward/backward passes through the learned feature map as the other methods, since both the data points and the inducing points must be processed. Thus, this choice is not disadvantageous to SVGP in terms of per-iteration feature-map computation. We do not claim that this allocation is optimal; a full sweep over the batch-size/inducing-point trade-off is beyond the scope of this work. The monotonicity of the optimized inducing-point variational bound gives qualitative guidance for the untested allocations: increasing the number of inducing variables cannot decrease the variational lower bound, and hence cannot increase the KL gap to the exact posterior, for fixed hyperparameters and exact optimization (Titsias, 2009, Sec. 3.1, Prop. 1). For our algorithms, the MINIMAX (Subsection 3.1.) and the SCGD (Subsection 3.2.), we approximated the Gaussian kernel $k'_u$ using the *Random Fourier Features* method with random features $\varphi$ of dimension 1000 such that

$$k'_u\left(z, z'\right) \approx \langle \varphi\left(z\right), \varphi\left(z'\right)\rangle.$$

We ran each test on five different splits of 90% for training and 10% for testing. We used AdaDelta for all methods, and for each combination of dataset, split, and method, we employed grid search to select the learning rate that achieves minimal marginal likelihood. For the SCGD algorithm, we fixed $b_t = 0.9$. Note that this is very close to 1, which means that SCGD becomes quite similar to BSGD, and much of the improvement comes just from the correct scaling of $\tilde{F}_t$. For MINIMAX, we fixed $\mu_t = 1.0$. We found this sufficient for achieving good results, despite the fact that, in theory, it should be increased in an outer loop. We ran each algorithm for 100 epochs and used the hyperparameters from the iteration in which the marginal likelihood achieved its minimum value.

When the assumed GP model is correctly specified, the hyperparameters are identifiable, and enough data are available, marginal-likelihood optimization is statistically aligned with prediction. However, in realistic settings, the assumed kernel family, feature map, noise model, or hyperparameterization may be misspecified. In that case, the hyperparameters that best fit the data under the assumed probabilistic GP model need not be the ones that minimize squared prediction error. Thus, a lower negative log marginal likelihood need not translate monotonically into a lower test RMSE. Because hyperparameters were selected by negative marginal likelihood rather than validation RMSE, the experimental protocol was not explicitly tuned for predictive squared error.

**Result with linear kernel - the exact case:** It seems (Table 2a and Figure 3) that the fact that our algorithm performs exact stochastic optimization brings a significant improvement over existing methods in optimizing the marginal likelihood. As expected, we can see that this advantage is more significant when the batch size is smaller.

**Result with Gaussian kernel:** Since our algorithms, MINIMAX and SCGD, use approximated Gaussian kernel based random Fourier features, we are no longer performing an exact optimization in this case. However, we can see in Table 2c and Figure 4 that, although we use an approximated kernel, when the restrictions on batch size are high, our methods perform better than the existing methods in optimizing the marginal likelihood. The advantages of our methods in optimization are also reflected in the test error (see Table 2d and Figure 6).

We see that in both cases, the finite dimensional RKHS and the infinite dimensional RKHS, our approach, unlike the existing inference algorithms, shows no degradation in the results of our algorithms with the decrease in batch size. This property can be vital for inference on weak edge devices where memory restrictions limit the possible batch size.

**Compute power used:** This experiment required roughly 20 days of g5.48xlarge machine time on AWS.

### 5.3 Attention-Based GP Regression

To better illustrate when our methods are most useful, we repeat the linear-kernel GP regression experiment on a subset of the UCI datasets, but replace the feature map used by the kernel. Concretely, as in the previous experiment we define a linear kernel on top of learned representations,

$$k(x, x') = \langle g_w(x),\, g_w(x') \rangle,$$

and we keep the GP model; the only change is that $g_w$ is now an attention-based network rather than the two-layer MLP used earlier. This setting is particularly relevant under memory constraints: the activation memory footprint of self-attention grows with the mini-batch size, making large batches substantially more expensive. We therefore compare how peak GPU memory increases with the batch size (Figure 9). In addition, to evaluate optimization speed, we plot the best (lowest) negative log marginal likelihood achieved up to a given training epoch (Figure 7) and up to a given wall-clock time (Figure 8). Overall, we observe that in this attention-based setting, SCGD in particular often provides a favorable trade-off between memory usage and convergence speed relative to the baselines.

In this experiment, we used the ADAM optimizer with a grid search of the learning rates.

**The network (attention feature map).** Let $p_{\text{in}}$ denote the number of input coordinates. The attention feature map $g_w : \mathbb{R}^{p_{\text{in}}} \to \mathbb{R}^d$ operates on the standardized input $x \in \mathbb{R}^{p_{\text{in}}}$ as a feature-wise Transformer encoder. We first apply a LayerNorm over the $p_{\text{in}}$ input coordinates and then tokenize each scalar feature into a $d$-dimensional token using a feature-specific affine embedding: for $j \in \{1, \ldots, p_{\text{in}}\}$,

$$t_j \;=\; x_j\, w_j + b_j \in \mathbb{R}^d,$$

where $w_j, b_j \in \mathbb{R}^d$ are learned parameters. We prepend a learned [CLS] token and feed the resulting sequence of length $p_{\text{in}} + 1$ through $L = 2$ Transformer encoder layers with model width $d = d_{\text{model}} = 32$, $H = 4$ attention heads per layer, and a position-wise feed-forward subnetwork of width $d_{\text{ff}} = 2d$ (i.e., mlp_ratio = 2). Each encoder layer uses a pre-LayerNorm (pre-norm) configuration and GELU nonlinearities. Dropout is disabled (dropout= 0) to keep the feature map deterministic. The final representation $g_w(x)$ is the LayerNorm-normalized [CLS] embedding after the last encoder layer. We do not use positional encodings or attention masks, since the tokens correspond to input dimensions rather than an ordered sequence. For initialization, the tokenizer weights are sampled with Xavier-uniform initialization, tokenizer biases are initialized to zero, and the [CLS] token is initialized from $\mathcal{N}(0, 0.02^2)$.

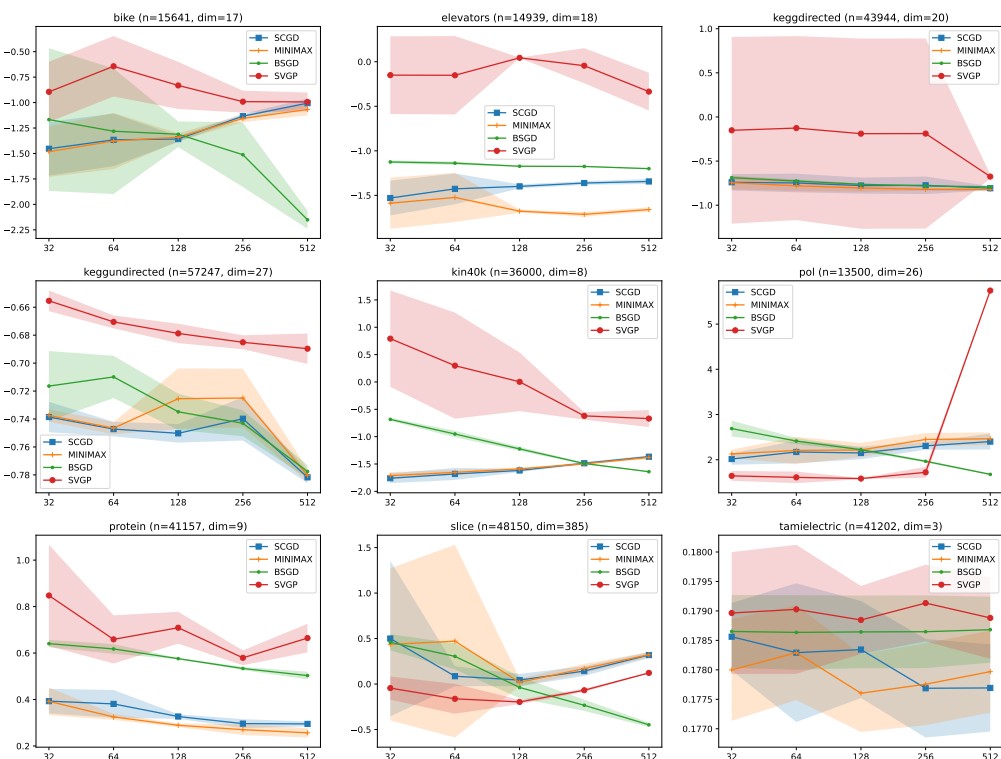

Figure 3: The negative marginal log likelihood divided by $n$ as a function of the batch size when using the

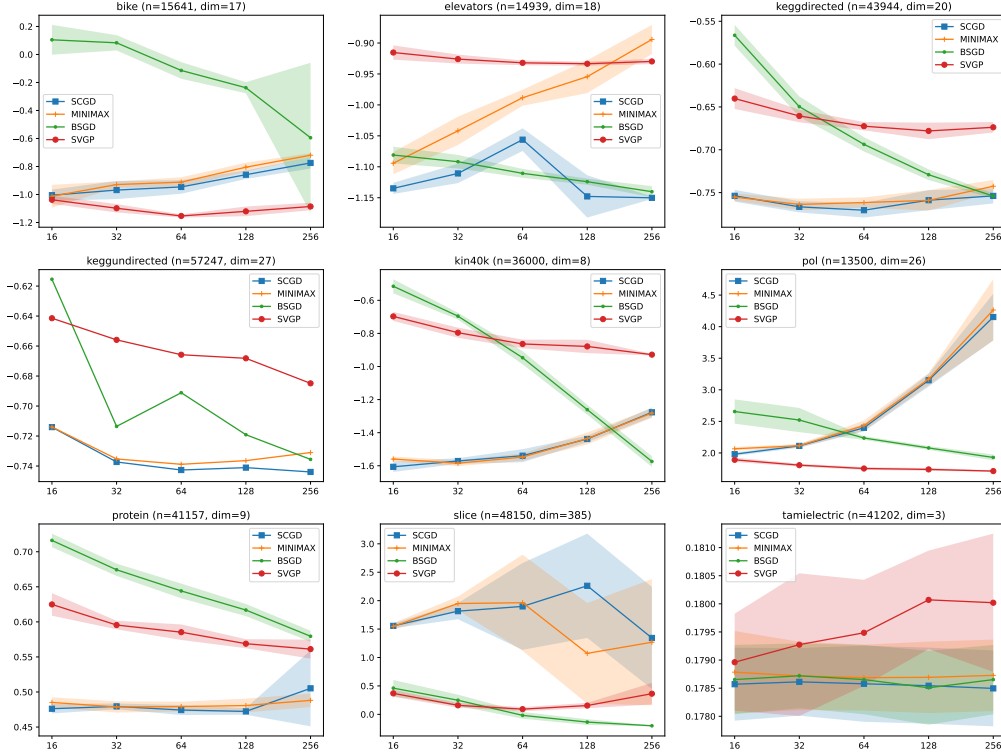

Figure 4: The negative marginal log likelihood divided by $n$ as a function of the batch size when using the Gaussian kernel, plotted with standard deviation margins.

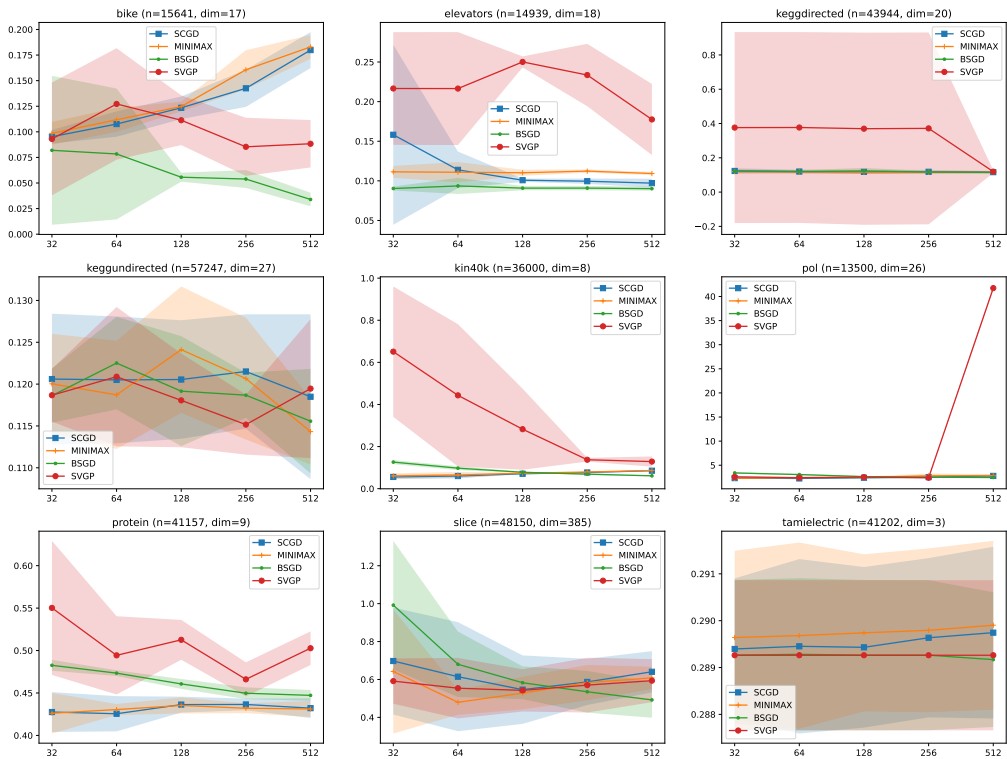

Figure 5: The RMSE as a function of the batch size when using the linear kernel, plotted with standard

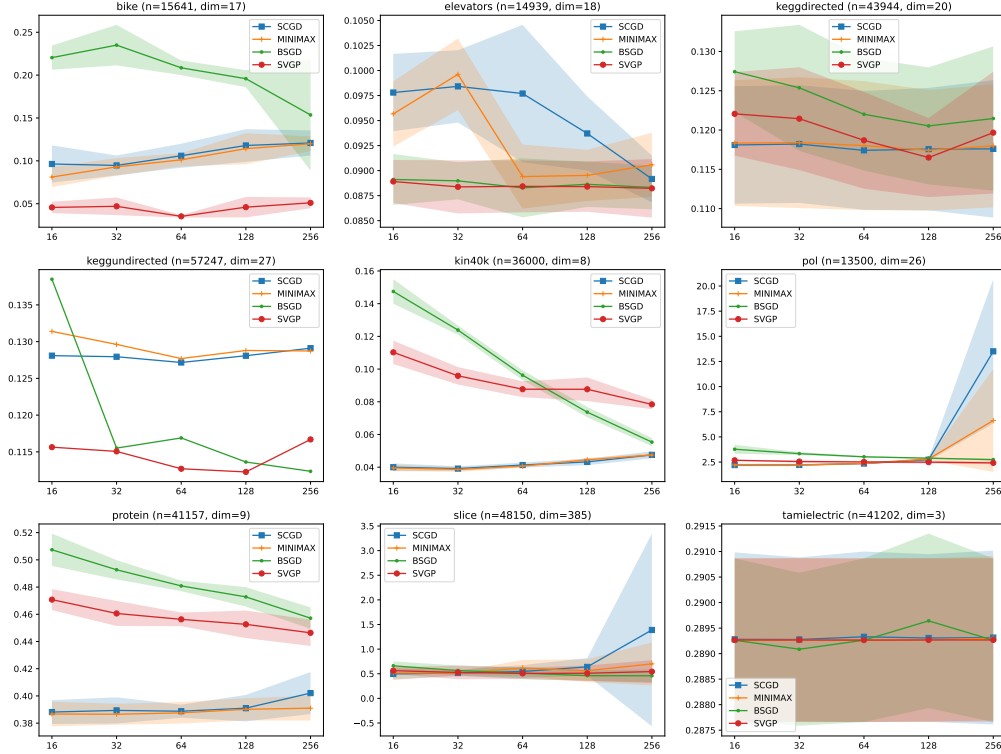

Figure 6: The RMSE as a function of the batch size when using the Gaussian kernel, plotted with standard deviation margins.

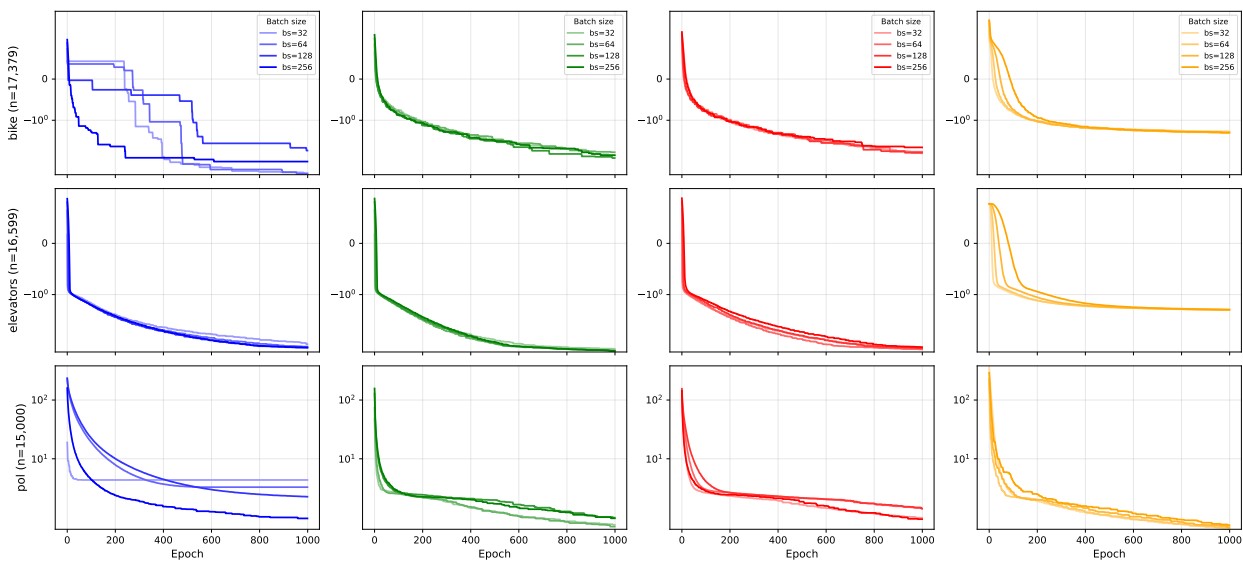

Figure 7: Negative log marginal likelihood as a function of training epoch.

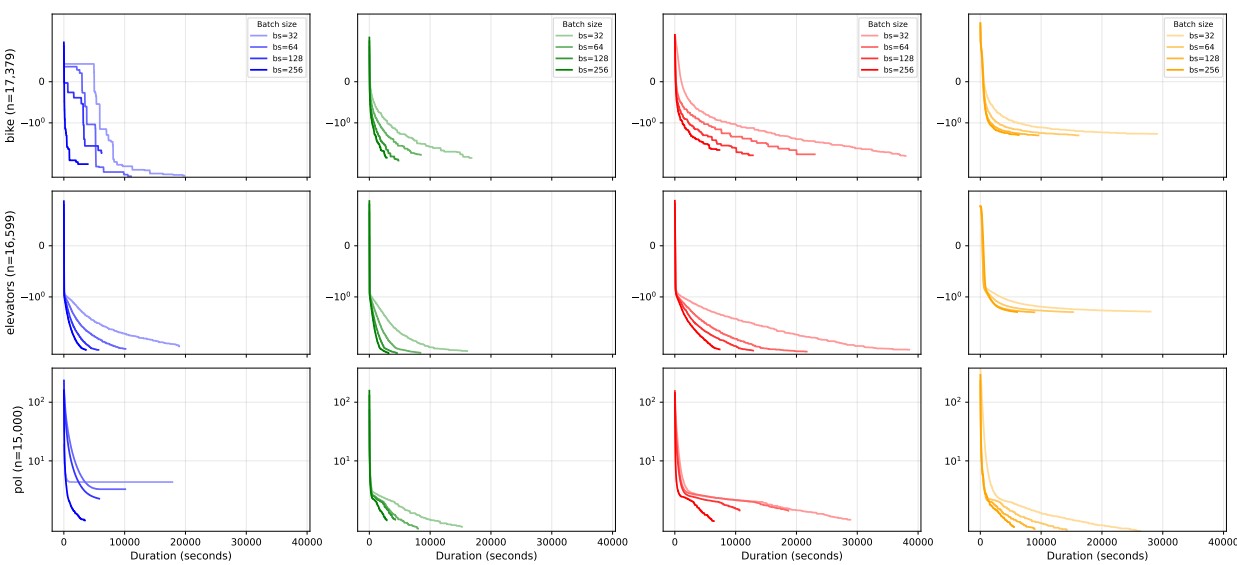

Figure 8: Negative log-likelihood as a function of duration.

## 6 Conclusion

In many cases, the covariance function of a GP is defined as an inner product between features of finite and moderate dimensions. In such cases, the problem of minimizing the negative-log-marginal-likelihood takes the shape of a standard ridge regression problem with a non-standard regularization term in the form of the log-determinant of the covariance matrix of the representations plus $\sigma^2 I$. In this work, we developed two techniques that enable the solution of this problem using stochastic mini-batches, which, unlike existing methods, do not depend on large batches to be exact. When inference involves forward and backward passes of a feedforward neural net, this property is of great importance and can enable such inference architectures on weak edge devices.

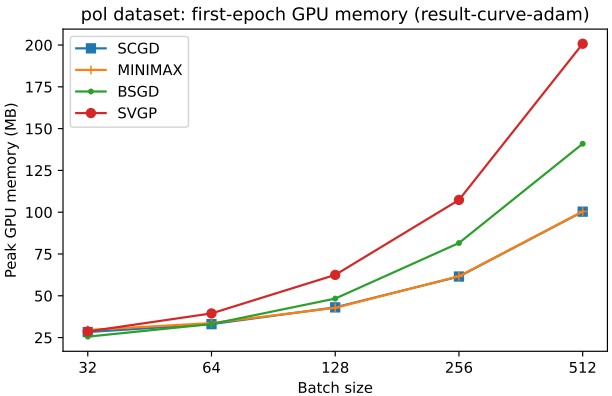

Figure 9: Peak GPU memory usage as a function of batch size.

**Limitations:** Both MINIMAX (Algorithm 1) and SCGD (Algorithm 2) maintain and repeatedly factorize a $d \times d$ matrix (either $A$ or $\tilde{F}_t$). This yields an additional per-iteration cost of $O(d^3)$ time (e.g., eigenvalue decomposition / Cholesky) and $O(d^2)$ memory, beyond the cost of computing $\phi_\alpha(x)$ and its Jacobian. Consequently, our methods are most attractive when the feature dimension $d = \dim\phi_\alpha(x)$ is *moderate* and when memory constraints force small mini-batches, but they can be less competitive when $d$ must be very large (e.g., when using a large number of random features to approximate an infinite-dimensional kernel). Our formal exactness guarantee should therefore be read as exactness for the finite-dimensional kernel actually optimized. Common kernels such as Gaussian and Matérn kernels on continuous domains generally do not admit a moderate exact finite-dimensional feature map. Although the kernel matrix on a fixed training set can be factorized exactly using at most $n$ features, this would make $d$ scale with $n$ and would reintroduce the $O(n^2)$ memory and $O(n^3)$ linear-algebra costs of full GP training. Thus, applying our framework to such kernels requires a finite-feature approximation. With Random Fourier Features (RFFs), the finite-sample terms decompose exactly for the RFF kernel, so the formal optimization guarantees are with respect to the finite-feature approximate GP, not the marginal likelihood of the original infinite-dimensional GP. The discrepancy depends on the quality of the finite-feature approximation on the training set, and for hyperparameter gradients also on the corresponding derivative approximation. Increasing the number of features reduces this approximation error but increases the $O(d^2)$ memory and $O(d^3)$ linear-algebra costs; if a very large number of features is required, the practical advantage of the method diminishes. The same caveat applies to data-dependent finite-feature approximations such as Nyström features (Williams & Seeger, 2001). If $U = \{u_1, \ldots, u_m\}$ is a fixed landmark set, then the Nyström approximation induces

$$\phi_\alpha^{\mathrm{Nys}}(x) = K_\alpha(U,U)^{-1/2} k_\alpha(U,x),$$

and therefore fits our framework with $d = m$. In this case, the per-sample quantities $g_i(\theta)$ and $F_i(\theta)$ still decompose over training examples, so mini-batch estimates remain unbiased for the Nyström-approximated objective. However, the exactness guarantee is then with respect to the fixed Nyström approximate kernel, not necessarily the original infinite-dimensional kernel. If landmarks are resampled, selected separately within each mini-batch, or adapted during optimization, the feature map and objective change over time, and our current analysis does not directly apply. Nyström features may reduce the required feature dimension when good landmark sets are available, but studying landmark selection and adaptive Nyström variants is beyond the scope of this work. In addition, the penalized formulation in MINIMAX introduces a trade-off through $\mu$: larger $\mu$ enforces $A \approx F(\theta)$ more strongly but can make the optimization increasingly ill-conditioned, while smaller $\mu$ improves numerical stability but may yield stationary points that are farther from those of the constrained objective. Finally, when resources allow large batches (or many inducing points), biased or variational baselines can often achieve comparable performance at lower linear-algebra cost, reducing the practical advantage of exact mini-batch gradients.

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

# A Complexity Analysis

We analyze the computational complexity of each of the algorithms. We report the linear-algebra terms separately from the feature-map costs accounted for in Table 1. We let $d$ be the dimension of $\phi_\alpha(x_i)$ and $b$ be the data mini-batch size.

## A.1 Minimax

**Computational Complexity Forward**

| Component | Complexity |
|---|---|
| $g_i$ | $O\left(d\right)$ |
| $F_i$ | $O\left(d^2\right)$ |
| $h\left(A\right)$ | $O\left(d^3\right)$ |
| $\frac{\left\langle B, \frac{1}{n}A - F_i(\theta)\right\rangle}{\|A\|}$ | $O\left(d^2\right)$ |
| **Total** | $O\left(d^3\right) + O\left(bd^2\right)$ |

**Computational Complexity Backward**

| Component | Complexity |
|---|---|
| $\frac{\partial g_i}{\partial w}$ | $O\left(d\right)$ |
| $\frac{\partial g_i}{\partial \phi_i}$ | $O\left(d\right)$ |
| $\frac{\partial h}{A} = A^{-1}$ | $O\left(d^3\right)$ |
| $\frac{\partial}{\partial B}\frac{\left\langle B, \frac{1}{n}A\right\rangle}{\|A\|} = \frac{\frac{1}{n}A}{\|A\|}$ | $O\left(d^2\right)$ |
| $\frac{\partial}{\partial A}\frac{\left\langle B, \frac{1}{n}A\right\rangle}{\|A\|} = \frac{\frac{1}{n}B\|A\| - \left\langle B, \frac{1}{n}A\right\rangle\frac{A}{\|A\|}}{\|A\|^2}$ | $O\left(d^2\right)$ |
| $\frac{\partial}{\partial \phi_i}\frac{\left\langle B, F_i\right\rangle}{\|A\|} = \frac{2B\phi_i}{\|A\|}$ | $O\left(d^2\right)$ |
| $proj_{\{\sigma^2 I_d \preceq M \preceq A_{\max}\}}(A)$ | $O(d^3)$ |
| **Total** | $O\left(d^3\right) + O\left(bd^2\right)$ |

**Memory Complexity Forward**

| Component | Complexity |
|---|---|
| $g_i$ | $O\left(d\right)$ |
| $A, B, F_i$ | $O\left(d^2\right)$ |
| **Total** | $O\left(d^2\right)$ |

**Memory Complexity Backward**

| Component | Complexity |
|---|---|
| $\frac{\partial g_i}{\partial w}$ | $O\left(d\right)$ |
| $\frac{\partial g_i}{\partial \phi_i}$ | $O\left(d\right)$ |
| $\frac{\partial h}{A} = A^{-1}$ | $O\left(d^2\right)$ |
| $\frac{\partial}{\partial B}\frac{\left\langle B, \frac{1}{n}A\right\rangle}{\|A\|} = \frac{\frac{1}{n}A}{\|A\|}$ | $O\left(d^2\right)$ |
| $\frac{\partial}{\partial A}\frac{\left\langle B, \frac{1}{n}A\right\rangle}{\|A\|} = \frac{\frac{1}{n}B\|A\| - \left\langle B, \frac{1}{n}A\right\rangle\frac{A}{\|A\|}}{\|A\|^2}$ | $O\left(d^2\right)$ |
| $\frac{\partial}{\partial \phi_i}\frac{\left\langle B, F_i\right\rangle}{\|A\|} = \frac{2B\phi_i}{\|A\|}$ | $O\left(d\right)$ |
| $proj_{\{\sigma^2 I_d \preceq M \preceq A_{\max}\}}(A)$ | $O(d^2)$ |
| **Total** | $O\left(d^2\right) + O\left(bd\right)$ |

### A.2 SCGD

**Computational Complexity Forward**

| Component | Complexity |
|:---:|:---:|
| $g_i$ | $O(d)$ |
| $\tilde{F}^{-1}$ | $O(d^3)$ |
| $F_i$ | $O(d^2)$ |
| **Total** | $O(d^3) + O(bd^2)$ |

**Computational Complexity Backward**

| Component | Complexity |
|:---:|:---:|
| $\frac{\partial g_i}{\partial w}$ | $O(d)$ |
| $\frac{\partial g_i}{\partial \phi_i}$ | $O(d)$ |
| $\frac{\partial}{\partial \phi_i} \left\langle \tilde{F}_t^{-1}, F_i \right\rangle = 2\phi_i \tilde{F}_t^{-1}$ | $O(d^2)$ |
| **Total** | $O(bd^2)$ |

**Memory Complexity Forward**

| Component | Complexity |
|:---:|:---:|
| $g_i$ | $O(d)$ |
| $F_i$ | $O(d^2)$ |
| $\tilde{F}$ | $O(d^2)$ |
| **Total** | $O(d^2)$ |

**Memory Complexity Backward**

| Component | Complexity |
|:---:|:---:|
| $\frac{\partial g_i}{\partial w}$ | $O(d)$ |
| $\frac{\partial g_i}{\partial \phi_i}$ | $O(d)$ |
| $\frac{\partial}{\partial \phi_i} \left\langle \tilde{F}_t^{-1}, F_i \right\rangle = 2\phi_i \tilde{F}_t^{-1}$ | $O(d)$ |
| **Total** | $O(bd)$ |

### A.3 BSGD

$$\frac{\partial}{\partial Z} \left[ \mathbf{y}^T \left( K + \sigma^2 I \right)^{-1} \mathbf{y} + \log \left| K + \sigma^2 I \right| \right] = -2 \left( K + \sigma^2 I \right)^{-1} \mathbf{y} \mathbf{y}^T \left( K + \sigma^2 I \right)^{-1} Z + 2 \left( K + \sigma^2 I \right)^{-1} Z$$

Computational complexity: $O(b^3) + O(b^2 d)$.

Memory complexity: $O(b^2) + O(bd)$

## B   Missing Proofs

**Theorem 1.** *For all $\lambda > 0$, $V \in \mathbb{R}^{n \times d}$, and $\mathbf{b} \in \mathbb{R}^n$ we have*

$$\mathbf{b}^T \left( VV^T + \lambda I \right)^{-1} \mathbf{b} = \min_{\mathbf{w}} \ \frac{1}{\lambda} \left\| V\mathbf{w} - \mathbf{b} \right\|^2 + \left\| \mathbf{w} \right\|^2 .$$

*Proof.* Let $f(\mathbf{w}) = \frac{1}{\lambda}\|V\mathbf{w} - \mathbf{b}\|^2 + \|\mathbf{w}\|^2$. Since $\lambda > 0$, $f$ is strongly convex and has a unique minimizer. The first-order condition gives

$$\nabla f(\mathbf{w}) = \frac{2}{\lambda}V^T(V\mathbf{w} - \mathbf{b}) + 2\mathbf{w} = 0 \quad \Longleftrightarrow \quad (V^TV + \lambda I_d)\mathbf{w} = V^T\mathbf{b},$$

hence

$$\hat{\mathbf{w}} = (V^TV + \lambda I_d)^{-1}V^T\mathbf{b}.$$

Now evaluate at $\hat{\mathbf{w}}$:

$$\begin{aligned}
f(\hat{\mathbf{w}}) &= \frac{1}{\lambda}\|V\hat{\mathbf{w}} - \mathbf{b}\|^2 + \|\hat{\mathbf{w}}\|^2 \\
&= \frac{1}{\lambda}\left(\hat{\mathbf{w}}^TV^TV\hat{\mathbf{w}} - 2\mathbf{b}^TV\hat{\mathbf{w}} + \mathbf{b}^T\mathbf{b} + \lambda\hat{\mathbf{w}}^T\hat{\mathbf{w}}\right) \\
&= \frac{1}{\lambda}\left(\mathbf{b}^T\mathbf{b} - \left(2\mathbf{b}^TV\hat{\mathbf{w}} - \hat{\mathbf{w}}^T(V^TV + \lambda I_d)\hat{\mathbf{w}}\right)\right).
\end{aligned}$$

Using $(V^TV + \lambda I_d)\hat{\mathbf{w}} = V^T\mathbf{b}$, we have $\hat{\mathbf{w}}^T(V^TV + \lambda I_d)\hat{\mathbf{w}} = \mathbf{b}^TV\hat{\mathbf{w}}$, so

$$f(\hat{\mathbf{w}}) = \frac{1}{\lambda}\left(\mathbf{b}^T\mathbf{b} - \mathbf{b}^TV\hat{\mathbf{w}}\right) = \frac{1}{\lambda}\mathbf{b}^T\left(I_n - V(V^TV + \lambda I_d)^{-1}V^T\right)\mathbf{b}.$$

Finally, by the matrix inversion lemma,

$$(VV^T + \lambda I_n)^{-1} = \frac{1}{\lambda}\left(I_n - V(V^TV + \lambda I_d)^{-1}V^T\right),$$

so $f(\hat{\mathbf{w}}) = \mathbf{b}^T(VV^T + \lambda I_n)^{-1}\mathbf{b}$. $\qquad\square$

## B.1 SCGD

**Stochastic oracle and the notation $\tilde{\nabla}$.** Following Wang et al. (2017), we consider compositional objectives $\ell(x) = (v \circ u)(x)$ with $u : \mathbb{R}^p \to \mathbb{R}^m$ and $v : \mathbb{R}^m \to \mathbb{R}$. We assume access to a sampling oracle that, given $x$, returns: (i) a random sample $u_\omega(x) \in \mathbb{R}^m$ with $\mathbf{E}[u_\omega(x)] = u(x)$; (ii) a random matrix $\tilde{\nabla}u_\omega(x) \in \mathbb{R}^{p \times m}$ whose $j$th column is a (stochastic) gradient/subgradient of the scalar map $x \mapsto [u_\omega(x)]_j$; and (iii) a (possibly stochastic) gradient $\nabla v_\nu(y)$ of the outer function. When $u_\omega$ is differentiable, we may take $\tilde{\nabla}u_\omega(x) = \nabla u_\omega(x)$. We also follow Wang et al. (2017) in the notation $\partial\ell(x)$: if $\ell$ is convex, $\partial\ell(x)$ denotes the convex subdifferential; if $\ell$ is differentiable (possibly nonconvex), we interpret $\partial\ell(x) = \{\nabla\ell(x)\}$.

**Assumption 1** (Wang et al., 2017, Assumption 1). *Let $\ell(x) = (v \circ u)(x)$, where $u : \mathbb{R}^p \to \mathbb{R}^m$ and $v : \mathbb{R}^m \to \mathbb{R}$. Let $C_u, C_v, V_u, L_v > 0$ be positive scalars.*

(i) *The outer function $v$ is continuously differentiable, the inner function $u$ is continuous, the feasible set $X \subseteq \mathbb{R}^p$ is closed and convex, and there exists at least one optimal solution $x^\star$ to the problem $\min_{x \in X} \ell(x)$.*

(ii) *The random variables $(\omega_0, \nu_0), (\omega_1, \nu_1), \ldots$ are i.i.d. with a complete probability measure, and*

$$\mathbf{E}[u_{\omega_0}(x)] = u(x), \qquad \mathbf{E}\left[\tilde{\nabla}u_{\omega_0}(x)\,\nabla v_{\nu_0}(u(x))\right] \in \partial\ell(x), \qquad \forall x \in X.$$

(iii) *The function $u(\cdot)$ is Lipschitz continuous with parameter $C_u$, and the samples $u_{\omega_k}(\cdot)$ and $\tilde{\nabla}u_{\omega_k}(\cdot)$ have bounded second moments such that, with probability 1,*

$$\mathbf{E}\left[\|\tilde{\nabla}u_{\omega_0}(x)\|^2 \mid \nu_0\right] \leq C_u, \qquad \mathbf{E}\left[\|u_{\omega_0}(x) - u(x)\|^2\right] \leq V_u, \qquad \forall x \in X.$$

(iv) *The functions $v$ and $v_\nu$ have Lipschitz continuous gradients such that, with probability 1,*

$$\mathbf{E}\left[\|\nabla v_{\nu_0}(y)\|^2\right] \leq C_v, \qquad \|\nabla v_{\nu_0}(y) - \nabla v_{\nu_0}(\bar{y})\| \leq L_v\|y - \bar{y}\|, \qquad \forall y, \bar{y} \in \mathbb{R}^m.$$

**Theorem 2** (Wang et al. 2017, Theorem 4: Convergence rate of basic SCGD for nonconvex problems).
*Assume Assumption 1 holds, $\ell$ is Lipschitz differentiable (i.e., $\nabla\ell$ is $L_\ell$-Lipschitz), and $X = \mathbb{R}^p$. Run basic SCGD (Algorithm 1 of Wang et al., 2017) with step sizes $\alpha_k = k^{-a}$ and $\beta_k = k^{-b}$ where $a, b \in (0, 1)$ satisfy $b < a < 2b$. Define*

$$T_\varepsilon := \min\Big\{k \geq 0 : \inf_{0 \leq t \leq k} \mathbf{E}\big[\|\nabla\ell(x_t)\|^2\big] \leq \varepsilon\Big\}.$$

*Then $T_\varepsilon = \mathcal{O}(\varepsilon^{-1/q})$, where $q = \min\{1-a,\ a-b,\ 2b-a,\ a\}$. In particular, $(a, b) = (\frac{3}{4}, \frac{1}{2})$ yields $T_\varepsilon = \mathcal{O}(\varepsilon^{-4})$.*

**Theorem 3** (SCGD for our GP objective). *Let $\theta = (\mathbf{w}, \alpha, \beta) \in \mathbb{R}^p$ and define*

$$\sigma^2(\beta) = \sigma_{\min}^2 + \exp(\beta) \quad (> 0).$$

*Recall*

$$\ell(\theta) = g(\theta) + \log\det(F(\theta)), \qquad F(\theta) = \sum_{i=1}^n u_{2,i}(\theta), \qquad g(\theta) = \sum_{i=1}^n u_{1,i}(\theta),$$

*where, for $i = 1, \ldots, n$,*

$$u_{1,i}(\theta) := \sigma(\beta)^{-2}\big(\phi_\alpha(x_i)^\top\mathbf{w} - y_i\big)^2 \;+\; \tfrac{1}{n}\|\mathbf{w}\|^2 \;+\; \tfrac{1}{n}(n - d)\log(\sigma^2(\beta)),$$
$$u_{2,i}(\theta) := \phi_\alpha(x_i)\phi_\alpha(x_i)^\top \;+\; \tfrac{1}{n}\sigma^2(\beta)\,I_d.$$

*Let $\text{vec}(\cdot)$ stack the entries of a matrix into a vector and let $\text{unvec}$ denote its inverse. Define the inner map*

$$u(\theta) = \big(u_1(\theta),\, \text{vec}(u_2(\theta))\big), \quad \text{where} \quad u_1(\theta) := \sum_{i=1}^n u_{1,i}(\theta), \; u_2(\theta) := \sum_{i=1}^n u_{2,i}(\theta),$$

*and the outer map*

$$v(s, z) = s + \log\det(\text{unvec}(z)), \qquad \text{dom}(v) = \{(s, z) : \text{unvec}(z) \succ 0\}.$$

*Then $\ell(\theta) = (v \circ u)(\theta)$ and $X = \mathbb{R}^p$.*

*Assume:*

(H1) *$x \mapsto \phi_\alpha(x)$ is $C^1$ in $\alpha$ and both $\phi_\alpha(x)$ and $\partial\phi_\alpha(x)/\partial\alpha$ are uniformly bounded on the (random) set of iterates $\{\theta_t\}_{t\geq0}$.*

(H2) *Mini-batches $\mathcal{S}_t$ are drawn i.i.d. uniformly from $\{1, \ldots, n\}$.*

*Run basic SCGD specialized to this composite:*

$$\tilde{u}_{2,t+1} = (1 - b_t)\tilde{u}_{2,t} \;+\; b_t\,\frac{n}{|\mathcal{S}_{t+1}|}\sum_{i \in \mathcal{S}_{t+1}} u_{2,i}(\theta_t),$$

$$\theta_{t+1} = \theta_t \;-\; a_t\,\frac{n}{|\mathcal{S}_{t+1}|}\sum_{i \in \mathcal{S}_{t+1}} \nabla_\theta\big[u_{1,i}(\theta_t) + \big\langle \tilde{u}_{2,t+1}^{-1},\, u_{2,i}(\theta_t)\big\rangle\big].$$

*With stepsizes $a_t = t^{-a}$ and $b_t = t^{-b}$ satisfying $b < a < 2b$, Assumption 1 holds for $(v, u)$ above by taking $\tilde{\nabla}u_\omega(\theta) = \nabla u_\omega(\theta)$ (Jacobian of the mini-batch inner map), and the nonconvex SCGD rate (Theorem 2) applies:*

$$\min_{0 \leq t \leq T} \mathbf{E}\big[\|\nabla\ell(\theta_t)\|^2\big] \;\leq\; C\,T^{-q}, \qquad q = \min\{\,1-a,\ a-b,\ 2b-a,\ a\,\}.$$

*In particular, with $(a, b) = (\frac{3}{4}, \frac{1}{2})$, the basic SCGD iterates reach an $\varepsilon$-stationary point in $\mathcal{O}(\varepsilon^{-4})$ iterations.*

*Proof.* We verify Assumption 1 for the pair $(v, u)$.

*A1(i): regularity and feasibility.* For any $\theta$,

$$u_2(\theta) = \sum_{i=1}^{n} \phi_\alpha(x_i)\phi_\alpha(x_i)^\top + \sigma^2(\beta)I_d \succeq \sigma^2(\beta)\, I_d \succeq \sigma_{\min}^2 I_d,$$

so $\log\det(\cdot)$ is $C^\infty$ on the range of $u_2$ and its matrix-gradient satisfies

$$\nabla_A \log\det(A) = A^{-\top} \quad (\text{and } = A^{-1} \text{ if } A \text{ is symmetric}).$$

Therefore $v(s, z) = s + \log\det(\mathrm{unvec}(z))$ is $C^1$ on $\mathrm{range}(u)$. Also, each $u_{1,i}(\theta)$ and $u_{2,i}(\theta)$ is continuous in $\theta$ under (H1) and the smooth map $\beta \mapsto \sigma^2(\beta)$, hence $u$ is continuous.

The feasible set in Theorem 3 is $X = \mathbb{R}^p$, which is closed and convex. Existence of an optimal solution is assumed as in Assumption 1(i).

*A1(ii): unbiased oracle and chain rule in expectation.* Let $\mathcal{S}$ be a uniform i.i.d. mini-batch and define the inner sample map

$$u_\omega(\theta) \equiv \tilde{u}(\theta; \mathcal{S}) = \left( \frac{n}{|\mathcal{S}|} \sum_{i\in\mathcal{S}} u_{1,i}(\theta),\ \mathrm{vec}\left( \frac{n}{|\mathcal{S}|} \sum_{i\in\mathcal{S}} u_{2,i}(\theta) \right) \right).$$

Then $\mathbf{E}[\tilde{u}(\theta; \mathcal{S})] = u(\theta)$ by uniform sampling.

**Definition of $\tilde\nabla$.** Because $\tilde{u}(\cdot; \mathcal{S})$ is differentiable in our setting, we take

$$\tilde\nabla u_\omega(\theta) := \nabla_\theta \tilde{u}(\theta; \mathcal{S}),$$

the Jacobian of the mini-batch inner map.

Now write $A = \mathrm{unvec}(z)$. Then

$$\nabla v(s, z) = \left( 1,\ \mathrm{vec}(A^{-\top}) \right),$$

and by the usual chain rule,

$$\nabla\ell(\theta) = \nabla_\theta u(\theta)\, \nabla v(u(\theta)).$$

Taking expectations yields

$$\mathbf{E}\big[\tilde\nabla u_\omega(\theta)\, \nabla v(u(\theta))\big] = \nabla_\theta u(\theta)\, \nabla v(u(\theta)) = \nabla\ell(\theta) \in \partial\ell(\theta),$$

which is exactly Assumption 1(ii) (and here the inclusion is equality since $\ell$ is smooth).

*A1(iii): Lipschitz $u$ and bounded second moments.* Under (H1), along the iterates the quantities $\phi_\alpha(x_i)$ and $\partial\phi_\alpha(x_i)/\partial\alpha$ are uniformly bounded, and $\sigma^2(\beta) \geq \sigma_{\min}^2$. Hence the Jacobians $\nabla_\theta u_{1,i}(\theta)$ and $\nabla_\theta u_{2,i}(\theta)$ are uniformly bounded along the iterates, which implies $u$ is Lipschitz on the set visited by $\{\theta_t\}$. Moreover, $\tilde{u}(\theta; \mathcal{S})$ and $\nabla_\theta\tilde{u}(\theta; \mathcal{S})$ are averages of bounded quantities, hence they have uniformly bounded second moments (in particular, the conditional moment bound in Assumption 1(iii) holds).

*A1(iv): Lipschitz $\nabla v$ on $\mathrm{range}(u)$.* Over $\mathrm{range}(u)$ we have $\lambda_{\min}(A) \geq \sigma_{\min}^2$. For any $A, B \succ 0$,

$$\|A^{-1} - B^{-1}\| = \|A^{-1}(B - A)B^{-1}\| \leq \|A^{-1}\|\, \|B^{-1}\|\, \|A - B\|.$$

Since $\|A^{-1}\| \leq 1/\sigma_{\min}^2$ and $\|B^{-1}\| \leq 1/\sigma_{\min}^2$ on the range,

$$\|A^{-1} - B^{-1}\| \leq \sigma_{\min}^{-4}\, \|A - B\|.$$

Thus $A \mapsto A^{-\top}$ is Lipschitz on $\mathrm{range}(u)$, and consequently $\nabla v$ is Lipschitz there.

Finally, $\nabla\ell$ is Lipschitz on the set visited by $\{\theta_t\}$ because it is the product/chain of bounded and Lipschitz maps $\nabla u$ and $\nabla v$ on that set. Therefore the conditions of Theorem 2 apply and the stated rate follows. $\qquad\square$

---

**Algorithm 3** Proximal alternating GDA *(Algorithm 2.1 in Boţ & Böhm, 2023)*

---

**Require:** Initial point $(x_0, y_0) \in \mathbb{R}^d \times \mathbb{R}^n$, stepsizes $\eta_x, \eta_y > 0$
1: **for** $k = 0, 1, 2, \ldots$ **do**
2: $\quad x_{k+1} \leftarrow \text{prox}_{\eta_x f}\big(x_k - \eta_x\, G_x(x_k, y_k)\big)$
3: $\quad y_{k+1} \leftarrow \text{prox}_{\eta_y h}\big(y_k + \eta_y\, G_y(x_{k+1}, y_k)\big)$
4: **end for**
$\quad$ Here $G_x$ and $G_y$ denote the appropriate (sub)gradients or stochastic (sub)gradient estimators in the deterministic or stochastic settings, respectively.

---

## B.2 Minimax

**Assumption 2** (Nonconvex–concave structure and smoothness (Assump. 3 in Boţ & Böhm, 2023)). *Let $\Phi : \mathbb{R}^p \times \mathbb{R}^q \to \mathbb{R}$ be the coupling term in the minimax template $\min_x \max_y \{f(x) + \Phi(x, y) - h(y)\}$.*

*(i) For every $x$, the map $y \mapsto \Phi(x, y)$ is concave and $L_{\nabla\Phi}$–smooth: $\|\nabla_y\Phi(x, y) - \nabla_y\Phi(x, y')\| \leq L_{\nabla\Phi}\|y - y'\|$.*

*(ii) For every $y$, the map $x \mapsto \Phi(x, y)$ is $\rho$–weakly convex, i.e., $x \mapsto \Phi(x, y) + \frac{\rho}{2}\|x\|^2$ is convex.*

**Assumption 3** (Lower bounded regularized max function (Assump. 4 in Boţ & Böhm, 2023)). *The regularized max function $\psi(x) := f(x) + \varphi(x)$ with $\varphi(x) := \max_y \{\Phi(x, y) - h(y)\}$ is bounded below: $\inf_x \psi(x) > -\infty$.*

**Assumption 4** (Lipschitz continuity in $x$ (Assump. 5 in Boţ & Böhm, 2023)). *There exists $L > 0$ such that for all $x, x'$ and all $y \in \text{dom}\, h$, $\big|\Phi(x, y) - \Phi(x', y)\big| \leq L\|x - x'\|$.*

**Assumption 5** (Convex regularizers and bounded dual domain (Assump. 6 in Boţ & Böhm, 2023)). *The functions $f$ and $h$ are proper, l.s.c., convex. Moreover, either $f$ is $L_f$–Lipschitz on its (open) domain or $f = \iota_{\mathcal{X}}$ for a nonempty closed convex set $\mathcal{X}$, and $\text{dom}\, h$ is bounded with diameter $D_h < \infty$.*

**Assumption 6** (Stochastic oracles: unbiasedness and bounded variance (Assump. 1–2 in Boţ & Böhm, 2023)). *The stochastic (sub)gradient estimators $G_x, G_y$ used in alternating GDA satisfy:*

*(i) Unbiasedness: $\mathbb{E}[G_x] = \nabla_x\Phi(x, y)$ and $\mathbb{E}[G_y] = \nabla_y\Phi(x, y)$.*

*(ii) Bounded variance: $\mathbb{E}\|G_x - \nabla_x\Phi(x, y)\|^2 \leq \sigma_{\text{stoch}}^2$ and $\mathbb{E}\|G_y - \nabla_y\Phi(x, y)\|^2 \leq \sigma_{\text{stoch}}^2$.*

**Indicator functions and projection as a proximal operator** For a (nonempty) set $\mathcal{C} \subseteq \mathbb{R}^p$, we use the extended–real valued *indicator* $\iota_{\mathcal{C}} : \mathbb{R}^p \to \mathbb{R} \cup \{+\infty\}$,

$$\iota_{\mathcal{C}}(z) := \begin{cases} 0, & z \in \mathcal{C}, \\ +\infty, & z \notin \mathcal{C}. \end{cases}$$

With this notation, $\text{prox}_{\eta f}(u) := \arg\min_z \{f(z) + \frac{1}{2\eta}\|z - u\|^2\}$ specializes to Euclidean projection when $f = \iota_{\mathcal{C}}$.

**Definition (Moreau–envelope $\varepsilon$–stationarity).** Let $\psi : \mathbb{R}^p \to \mathbb{R} \cup \{+\infty\}$ be proper, l.s.c., and $\rho$–weakly convex, and fix any $\lambda \in (0, \rho^{-1})$. We say that $x$ is $\varepsilon$–*stationary for $\psi$ in the Moreau sense* if

$$\big\|\nabla\psi_\lambda(x)\big\| \leq \varepsilon, \quad \text{where} \quad \psi_\lambda(x) = \min_z \Big\{\psi(z) + \frac{1}{2\lambda}\|z - x\|^2\Big\}.$$

Equivalently, with $\hat{x} := \text{prox}_{\lambda\psi}(x)$, we have $\nabla\psi_\lambda(x) = \lambda^{-1}(x - \hat{x})$ and hence $\|x - \hat{x}\| \leq \lambda\varepsilon$.

**Lemma 1** (Boţ & Böhm 2023, Lemma 2.3). *Let $x$ be $\varepsilon$-stationary for the proper, $\rho$-weakly convex and l.s.c. function $\psi$, i.e., $\|\nabla\psi_\lambda(x)\| \leq \varepsilon$ with $\lambda \in (0, \rho^{-1})$. Then there exist a point $\hat{x}$ such that $\|x - \hat{x}\| \leq \varepsilon\lambda$ and $\text{dist}\big(0, \partial\psi(\hat{x})\big) \leq \varepsilon$.*

**Theorem 4** (Stochastic alternating prox–GDA for nonconvex–concave problems). (Boţ & Böhm 2023, Theorem 3.2). *Under Assumptions 2–6, consider Algorithm 3. Let $\phi(x) = \max_y\{\Phi(x,y) - h(y)\}$ and $\psi(x) = f(x) + \phi(x)$. For any $\lambda \in (0, \rho^{-1})$, the Moreau envelope $\psi_\lambda$ is $C^1$, and the iterates satisfy*

$$\min_{0 \leq t \leq K} \mathbb{E}\big[\|\nabla \psi_\lambda(x_t)\|\big] \leq \varepsilon \quad whenever \quad K = \mathcal{O}\big(\varepsilon^{-8}\big),$$

*with constants depending only on problem Lipschitz/variance parameters and diameters of the constraint sets.*

**Mapping to the Boţ & Böhm (2023) template.** Set the primal variable as $x \equiv \zeta = (\theta, A)$ with $\theta = (\mathbf{w}, \alpha, \beta)$ and $\sigma^2 = \sigma_{\min}^2 + e^\beta$, and the dual variable as $y \equiv B \in \mathbb{R}^{d \times d}$. Define the convex constraint sets

$$\mathcal{X} \equiv \Omega_1 := \big\{\, (\theta, A) : \; \theta_{\min} \preceq \theta \preceq \theta_{\max}, \;\; \sigma^2 I_d \preceq A \preceq A_{\max} \,\big\}, \qquad \mathcal{Y} \equiv \Omega_2 := \big\{\, B : \; \|B\| \leq 1 \,\big\},$$

and take $f = \iota_{\Omega_1}$, $h = \iota_{\Omega_2}$. Let

$$\Phi(\zeta, B) := g(\theta) + \log|A| \; + \; \mu\, \frac{\langle B,\; A - F(\theta)\rangle}{\|A\|},$$

so that

$$\phi(\zeta) = \max_{B \in \Omega_2} \Phi(\zeta, B) = g(\theta) + \log|A| + \mu\, \frac{\|A - F(\theta)\|}{\|A\|}.$$

Hence the regularized max-function is $\psi(\zeta) = \iota_{\Omega_1}(\zeta) + \phi(\zeta) = \iota_{\Omega_1}(\zeta) + \ell_\mu(\zeta)$, where $\ell_\mu(\zeta)$ is precisely our penalized objective in Section 3.1. With this identification, Algorithm 1 is exactly the projected alternating GDA of Boţ & Böhm (2023) applied to $f + \Phi - h$.

**Constraint-set properties.** The constraint sets $\Omega_1$ and $\Omega_2$ defined above satisfy:

$\Omega_1$ is nonempty, compact, and convex, and every $(\theta, A) \in \Omega_1$ satisfies
$$A \succeq \sigma^2 I_d, \qquad \sigma^2 \in [\sigma_{\min}^2, \sigma_{\max}^2], \tag{H1}$$
$\Omega_2 = \{B : \|B\| \leq 1\}$ is nonempty, compact, and convex.

**Theorem 5** (Convergence of Algorithm 1). *Let $\Omega_1$ and $\Omega_2$ be the constraint sets defined above, and set $\psi(\zeta) = \iota_{\Omega_1}(\zeta) + \ell_\mu(\zeta)$. Assume (H1) and:*

(H2) *The feature map $x \mapsto \phi_\alpha(x)$ is $C^1$ in $\alpha$, and both $\phi_\alpha(x)$ and $\partial \phi_\alpha(x)/\partial \alpha$ are uniformly bounded on $\Omega_1$ for all training points.*

(H3) *Mini-batches are sampled i.i.d. (uniformly) and the resulting stochastic gradients used in the two projected steps are unbiased with uniformly bounded second moments.*

*Then there exists $\rho > 0$ such that, for any $\lambda \in (0, \rho^{-1})$, the Moreau envelope $\psi_\lambda$ of $\psi(\zeta) = \iota_{\Omega_1}(\zeta) + \ell_\mu(\zeta)$ is $C^1$, and the iterates of Algorithm 1 with constant stepsizes chosen as in Boţ & Böhm (2023, Thm. 3.2) satisfy*

$$\min_{0 \leq t \leq K} \mathbb{E}\big[\|\nabla \psi_\lambda(\zeta_t)\|\big] \leq \varepsilon \qquad whenever \qquad K = \mathcal{O}\big(\varepsilon^{-8}\big).$$

*Equivalently, Algorithm 1 reaches an $\varepsilon$–stationary point of the penalized objective $\ell_\mu$ (in the Moreau–envelope sense) within $\mathcal{O}(\varepsilon^{-8})$ stochastic gradient evaluations.*

*Proof (verification of the reused assumptions).* Under the mapping $x \equiv \zeta$, $y \equiv B$ described above, we check the items required by Assumptions 2–6 of Boţ & Böhm (2023) for our choices of $\Phi$, $f = \iota_{\Omega_1}$ and $h = \iota_{\Omega_2}$.

*A2(i):* For fixed $\zeta$, $B \mapsto \Phi(\zeta, B)$ is affine:

$$\nabla_B \Phi(\zeta, B) = \mu\, \frac{A - F(\theta)}{\|A\|},$$

which is independent of $B$. Hence it is concave and $L_{\nabla\Phi} = 0$–smooth on $\Omega_2$.

*A2(ii)*: On $\Omega_1$ the functions $\theta \mapsto g(\theta)$ and $A \mapsto \log|A|$ are $C^1$ with Lipschitz gradients: $A \succeq \sigma_{\min}^2 I$ bounds $\|A^{-1}\|$, and (H2) bounds derivatives of $F(\theta)$. The penalty coupling

$$\zeta \mapsto \frac{\langle B,\, A - F(\theta) \rangle}{\|A\|}$$

is $C^1$ on $\Omega_1$ because $\|A\| \geq \|A\|_2 \geq \sigma_{\min}^2 > 0$, and its gradient is Lipschitz on the compact set $\Omega_1 \times \Omega_2$ (all ingredients $A, F(\theta), B$ are uniformly bounded by (H1)–(H2)). Therefore $\nabla_\zeta \Phi$ is globally Lipschitz on $\Omega_1 \times \Omega_2$ with some constant $L_\zeta$, so $x \mapsto \Phi(x, y)$ is $\rho$–weakly convex with $\rho := L_\zeta$.

*A3*: By (H1), $\Omega_1$ is bounded and $A \succeq \sigma_{\min}^2 I$; thus $\log|A|$ is bounded below, $g(\theta)$ is bounded below, and $\|A - F(\theta)\|/\|A\|$ is bounded. Hence $\psi(\zeta) = \iota_{\Omega_1}(\zeta) + \ell_\mu(\zeta)$ is bounded below.

*A4*: On $\Omega_1 \times \Omega_2$ we already observed $\nabla_\zeta \Phi$ is bounded; hence $|\Phi(\zeta, B) - \Phi(\zeta', B)| \leq L\|\zeta - \zeta'\|$ with some $L < \infty$ independent of $B \in \Omega_2$.

*A5*: $f = \iota_{\Omega_1}$ and $h = \iota_{\Omega_2}$ are proper, l.s.c., convex. The domain of $h$ equals $\Omega_2$, whose diameter is finite.

*A6*: By mini-batch sampling (H3) and linearity of expectation, the stochastic gradients used in the two projected steps are unbiased. Uniform boundedness of gradients on $\Omega_1 \times \Omega_2$ implies a finite variance bound $\sigma^2$.

Thus Assumptions 2–6 hold for our problem data under the above mapping to the template of Boţ & Böhm (2023). Applying Theorem 4 (with $x \equiv \zeta$, $y \equiv B$) yields the claimed $\mathcal{O}(\varepsilon^{-8})$ complexity for $\min_{t \leq K} \mathbb{E}\|\nabla\psi_\lambda(\zeta_t)\|$. Since $\psi = \iota_{\Omega_1} + \ell_\mu$, this is precisely stationarity of the penalized objective $\ell_\mu$ in the Moreau–envelope sense. $\qquad\square$

