# OpenReview forum: "Unbiased Stochastic Optimization for Gaussian Processes on Finite Dimensional RKHS"
_TMLR — Decision pending for TMLR_

### Review · Reviewer_4ouQ · 2026-04-08

**Summary Of Contributions:**

This paper tackles hyperparameter learning in Gaussian Processes (GPs), i.e. the minimisation of the negative marginal log-likelihood. More specifically, it aims at overcoming non-convergence or computational burdens issues related to this optimisation scheme. Hence, the authors here focus on the case where the kernel or covariance function at hand has a finite-dimensional feature map of moderate dimension or can be approximated by such a feature map (e.g. using Random Fourier Features (RFF)). Then, they provide the following contributions:
1. They propose a first algorithm by reframing the optimisation problem as a nonconvex-concave minimax problem (Minimax algorithm).
2. By writing the loss function as the composition of a function and the expected value of another function, they propose a second algorithm using Stochastic Compositional Gradient Descent (SCGD) method (SGCD algorithm)
3. They provide experiments on synthetic and real-world datasets and show the superiority of their method in the case where memory resources limit the feasible batch size and the possible number of inducing points. More specifically, in the finite-dimensional settings, their methods outperform existing ones with moderate batch size or number of inducing points, in the infinite-dimensional one, more severe limitations on the batch size and number of inducing points are needed to maintain this performance.
4. They provide theoretical convergence guarantees for the proposed algorithms.

Strengths:
- The paper is well-written.
- The paper is clearly motivated and the related works are discussed.
- The claims are supported by theoretical and empirical evidences.
- The authors discuss the limitations of their algorithms (about the dimension $d$, the trade-off through $\mu$ and computational resources).

Weaknesses:
- It would be better to have the theoretical guarantees in the main paper instead of the appendix.
- The scope of the paper might be a little narrow since they focus on the specific finite-dimensional setting.

**Audience:**

Yes

**Audience Explanation:**

I think it would interest the TMLR's audience as GPs are a very famous Bayesian framework for Machine Learning tasks, and more specifically, hyperparameter learning is one of their crucial questions.

**Claims And Evidence:**

Yes

**Claims Explanation:**

The claims made are supported with empirical and theoretical evidences, even though the theoretical evidences would deserve to be highlighted in the main paper instead of the appendix. Moreover, the authors discuss their methods with respect to related works and deal with the limitations implied by them. As an example, they provide a complexity analysis, which is of high-interest in the context of GPs and hyperparameter learning.

**Requested Changes:**

My main questions would be about the settings of the paper.

1. Infinite-dimension: Have you considered using data-dependent feature approximation such as Nystrom approximation (see e.g. [1]) instead of data-independent like RFF? Would there be an issue since you are doing batch-partitioning? Would it allow you to reduce the feature size further in some cases?

2. Computational resources: Have you tried comparing the computational times and energy consumption induced by your methods and standard benchmarks on both low and high computational resources? Would your methods fit in the "frugal AI" field?

It is not a mandatory requested change, but I would recommend moving the theoretical guarantees into the main paper and reorganizing a little the paper to not make it too long, such as moving some figures into the appendix.

---

> ### Author Response · Authors · 2026-05-06
>
> We thank the reviewer for the positive assessment and for the helpful question about data-dependent feature approximations.
>
> Fixed Nyström features do fit our framework. If a landmark set $U$ is selected before optimization and then kept fixed, the Nyström approximation induces the finite-dimensional feature map $\phi_{\alpha}^{\mathrm{Nys}}(x)=K_{\alpha}(U,U)^{-1/2}k_{\alpha}(U,x)$, so the approximate kernel has the finite-feature form required by our algorithms, with $d=m$. In this case, the per-sample quantities $g_i(\theta)$ and $F_i(\theta)$ still decompose over training examples, so mini-batch estimates remain unbiased for the Nyström-approximated objective.
>
> The important caveat is that the exactness guarantee is then with respect to the fixed Nyström approximate kernel, not the original infinite-dimensional kernel. If landmarks are resampled per batch or adapted during optimization, the feature map and objective change over time, and our current analysis does not directly apply. Nyström features may indeed reduce the required feature dimension when good landmarks are available. We have clarified this point in Section 2.2 and in the limitations.

---

### Review · Reviewer_Yd7w · 2026-04-11

**Summary Of Contributions:**

The paper proposes two stochastic optimization algorithms (MINIMAX and SCGD) for GP hyperparameter learning that, unlike existing methods (BSGD and SVGP), converge to a stationary point of the true marginal likelihood for any batch size without further approximations. The key challenge they address is that the log-determinant term in the GP marginal likelihood cannot be decomposed over mini-batches; MINIMAX handles this by recasting the problem as a nonconvex-concave minimax optimization, while SCGD treats it as a compositional objective and tracks the gram matrix via exponential smoothing. Both methods are advantageous when memory constraints force batch size to be small, a regime where BSGD degrades due to gradient bias and SVGP degrades due to an insufficient number of inducing points.

**Audience:**

Yes

**Audience Explanation:**

The problem of scaling GP hyperparameter learning to large datasets is well-known and practically important, and the specific angle of enabling exact stochastic optimization for finite-dimensional RKHS kernels is a meaningful contribution. Researchers working on GP scalability, deep kernel learning, or deployment of probabilistic models on memory-constrained devices would find the SCGD/MINIMAX formulations of interest, regardless of the mixed experimental results.

**Claims And Evidence:**

No

**Claims Explanation:**

While the synthetic recovery experiment (Figure 1) cleanly supports the strength of the proposed algorithms,  the pie charts (Figures 3–6) are visually ambiguous and hard to interpret. The authors acknowledge that better marginal likelihood does not consistently translate to better test RMSE (Figures 3, 5), attributing this to suboptimal learning rate selection, plausible but unsatisfying.  Overall, the marginal likelihood results in Figure 7 are the strongest evidence for the paper's claims, but the RMSE results and ambiguous aggregate figures (Figures 3-6) leave the practical advantage insufficiently established.

**Requested Changes:**

The following adjustments would be beneficial to strengthen the manuscript before acceptance. First, Figures 3–6 would benefit from a clearer presentation. A win/loss/tie table would make it easier to verify the claimed marginal likelihood advantage. Second, the relationship between marginal likelihood and RMSE results deserves a more careful explanation; attributing the discrepancy to suboptimal learning rates is plausible but would be more convincing with some empirical validation. Third, the SVGP comparison would be strengthened by either exploring the inducing-point/batch-size trade-off or providing a clearer argument for why constraining inducing points to equal the batch size constitutes a fair comparison.

The following adjustments would further strengthen the work. A discussion of the fundamental obstacles to extending the framework beyond inner-product kernels to commonly used kernels such as Matern would significantly broaden the paper's appeal. The limitations of the RFF-based extension to infinite-dimensional kernels should also be discussed more explicitly, particularly regarding how much the exactness guarantee is compromised in practice.

---

> ### Author Response · Authors · 2026-05-06
>
> Thank you for the constructive feedback. We revised the manuscript accordingly.
>
> First, we replaced the ambiguous aggregate pie-chart evidence with explicit win/loss count tables across datasets, kernels, batch sizes, and metrics, making the marginal-likelihood advantage easier to verify directly.
>
> Second, we strengthened the discussion of marginal likelihood versus RMSE. Our objective is exact marginal-likelihood optimization, and under model misspecification the hyperparameters preferred by marginal likelihood need not minimize squared prediction error.
>
> Third, we clarified the SVGP comparison. Our experiments partially sample the broader mini-batch/inducing-point allocation space by using the balanced allocation where the number of inducing points equals the mini-batch size. This is a fair fixed-budget point because both data points and inducing points require forward/backward passes through the learned feature map. For allocations not explicitly sampled, monotonicity of the optimized inducing-point ELBO gives practical directional guidance: increasing the number of inducing variables cannot decrease the optimized variational lower bound, and hence cannot increase the KL gap to the exact posterior, for fixed hyperparameters and exact optimization. This is only partial guidance, and a full study of this allocation space is beyond the scope of this work.
>
> Finally, we expanded the limitations to clarify that exactness applies to the finite-dimensional kernel actually optimized. For Gaussian/Matérn kernels, RFF or other finite-feature approximations make the optimization exact only for the approximate GP, not for the original infinite-dimensional GP.

---

### Review · Reviewer_k6J5 · 2026-04-30

**Summary Of Contributions:**

This paper studies stochastic hyperparameter learning for Gaussian processes in the setting where the kernel admits a finite-dimensional feature representation. The paper observes that naive mini-batching results in biased gradient estimates due to the log det term. The paper proposes two stochastic optimization methods to address this issue.

**Strengths**

1. The paper identifies a clear and important issue in stochastic GP hyperparameter learning: the natural mini-batch GP marginal likelihood is computationally convenient but does not provide unbiased gradients for the full marginal likelihood.

2. The paper provides two alternative approaches and provides both extensive theoretical results (convergence results, complexity analysis) and extensive experimentation.

3. The paper is very clear and up front about its limitations and I view this is a strength.

**Weakness**

This is minor but I would appreciate if the authors mentioned earlier that the naive method results in biased estimates.

**Audience:**

Yes

**Audience Explanation:**

GP regression is a topic of interest.

**Claims And Evidence:**

Yes

**Claims Explanation:**

Yes the theorems and experiments clearly support the claims in the paper. The small batch experiments show that the proposed method returns in better marginal likelihood.

As batch size increases, the bias reduces and other methods get more competitive.

**Requested Changes:**

This is not a requested change but a question.

The papers note that better marginal likelihood does not translate to better RMSE. Could this suggest that the bias in the biased estimates is helpful? Something like an implicit bias result?

---

> ### Author Response · Authors · 2026-05-06
>
> We thank the reviewer for the positive and constructive feedback.
>
> We agree that the biased nature of the naïve mini-batch method should be made explicit earlier. In the current version, we mention that the direct stochastic-batch approach gives biased gradient estimates, but we agree that the reason should be stated already when the log-determinant obstacle is introduced. We revised the Introduction to clarify that the log-determinant term is global and does not decompose over samples; therefore, replacing the full dataset by a mini-batch generally yields biased gradients of the full marginal likelihood.
>
> Regarding the RMSE question, we agree that this is an interesting possibility. Our interpretation is that cases where biased methods obtain better RMSE may be related to model misspecification, possibly combined with overfitting to the assumed GP model when optimizing the exact marginal likelihood.

---

### Decision · Action_Editor_jaSv · 2026-06-23

**Recommendation:** Accept as is

**Additional Comments:**

The complexity analysis in Section 4.1 is helpful for readers to understand the strength, limitations, and practical use case of the proposed algorithm. However, this comparison is only per-iteration. The authors should also discuss a little bit about the total complexity comparison incorporating the convergence speed (or the quality of the convergence) of different algorithms.

**Audience:**

Yes

**Audience Explanation:**

The paper solves a practical problem of the training of stochastic Gaussian Process Regression algorithms. The paper comes with formal convergence guarantees and convincing empirical experimental results. The problem and results are both of interest to the machine learning community.

**Claims And Evidence:**

Yes

**Claims Explanation:**

The paper provide formal convergence guarantees, sound theoretical proofs, empirical evaluations, and detailed discussion on the limitation of the proposed algorithms.